# IsoCompute Playbook: Optimally Scaling Sampling Compute for LLM RL

**Zhoujun Cheng** [* 1 2]  **Yutao Xie** [* 1]  **Yuxiao Qu** [* 3]  **Amrith Setlur** [* 3]  **Shibo Hao** [1 2]  **Varad Pimpalkhute** [2]
**Tongtong Liang** [1]  **Feng Yao** [1]  **Zhengzhong Liu** [2]  **Eric Xing** [2 3]  **Virginia Smith** [3]  **Ruslan Salakhutdinov** [3]
**Zhiting Hu** [1]  **Taylor Killian** [2]  **Aviral Kumar** [3]

## Abstract

While scaling laws guide compute allocation for LLM pre-training, analogous prescriptions for reinforcement learning (RL) post-training of LLMs remain poorly understood. We study the compute-optimal allocation of sampling compute for on-policy RL methods in LLMs, framing scaling as a compute-constrained optimization over three resources: parallel rollouts per problem, number of problems per batch, and number of update steps. We find that the compute-optimal number of parallel rollouts per problem increases predictably with compute budget and then saturates. This trend holds across both easy and hard problems, though driven by different mechanisms: solution sharpening on easy problems and coverage expansion on hard problems. We further show that increasing the number of parallel rollouts mitigates interference across problems, while the number of problems per batch primarily affects training stability and can be chosen within a broad range. Validated across base models and data distributions, our results recast RL scaling laws as prescriptive allocation rules and provide practical guidance for compute-efficient LLM RL post-training.

## 1. Introduction

A blocker to scaling up reinforcement learning (RL) for large language models (LLMs) is the absence of a concrete workflow: a recipe that tells practitioners *what* to scale, *how* to scale it, and *what outcomes of scaling* one should expect. In many areas of modern AI, such workflows are enabled by scaling laws (Hestness et al., 2017; Kaplan et al., 2020; Hoffmann et al., 2022), where initial experiments

reveal predictable relationships between performance and resources (e.g., compute, data). These laws guide compute allocation, model selection, and hyperparameter choices. In this paper, our goal is to understand and build analogous scaling laws for RL post-training of LLMs.

In contrast to pre-training or supervised learning, scaling behavior in RL is far less understood due to the tight coupling between **exploration** (data collection) and **optimization** (learning from data). Recent work has begun to characterize scaling behavior in classical deep RL (Jones, 2021; Hilton et al., 2023; Fu et al., 2025). However, in the LLM setting, this line of study remains in its infancy. The most relevant prior results show that, under a given fixed problem mixture, RL reward curves exhibit clean sigmoidal behavior when trained for longer (Khatri et al., 2025), or that RL performance scales with model size in a manner reminiscent of pre-training (Tan et al., 2025). While informative, these results stop short of addressing the central question that often plagues practitioners running RL: *how to allocate resources when setting up an RL run for a base model?* Given a base model, a problem distribution, and a fixed compute budget, how should one spend this compute to maximize downstream performance?

We address this question by studying the optimal allocation of **sampling compute** in LLM reinforcement learning. To this end, we conduct a series of experiments across three base models (Qwen2.5-7B-Instruct, Qwen3-4B-Instruct, and Llama 3.1-8B-Instruct), covering diverse training configurations and problem distributions, including easy, hard, and skewed mixtures of prompts. Our analysis reveals a nuanced picture. Unlike pre-training, scaling behavior in RL is governed not only by total compute, but also by its interaction with the base model and the prompt distribution. Nevertheless, under *healthy* and stable training recipes, we find that predictable allocation rules emerge for key hyperparameters as a function of sampling compute. Concretely, for on-policy RL methods that optimize LLM policies using multiple parallel rollouts per sequential gradient step, we make the following observations as in Figure 1, validated across about $120,000$ GPU-hours of RL experiments.

***First***, the compute-optimal number of parallel rollouts per

---

[1]UC San Diego [2]MBZUAI-IFM [3]Carnegie Mellon University. Correspondence to: Zhoujun Cheng <z6cheng@ucsd.edu>, Yutao Xie <yux076@ucsd.edu>, Yuxiao Qu <yuxiaoq@andrew.cmu.edu>, Amrith Setlur <asetlur@andrew.cmu.edu>.

*Proceedings of the $43^{rd}$ International Conference on Machine Learning*, Seoul, South Korea. PMLR 306, 2026. Copyright 2026 by the author(s).

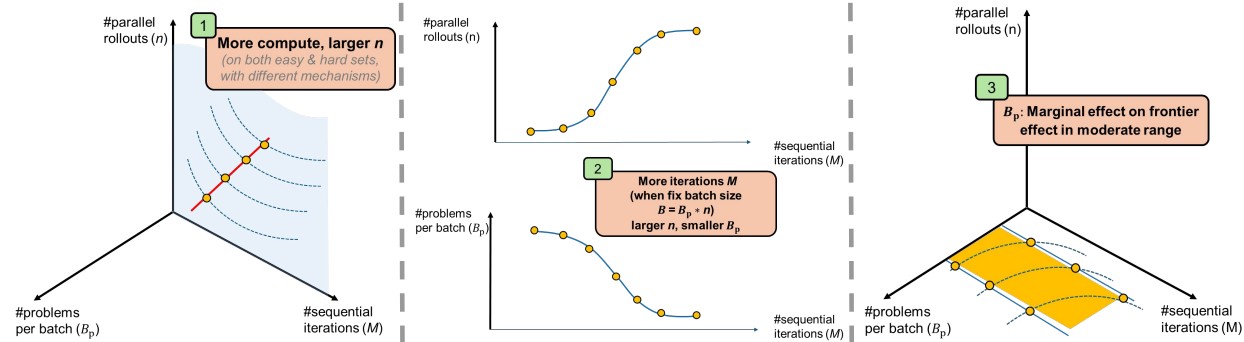

*Figure 1.* **Compute-optimal sampling laws for LLM RL.** We study allocation of sampling compute along three axes: number of parallel rollouts per problem ($n$), number of problems per batch ($B_p$), and number of sequential iterations ($M$), where the total compute is $C = B_p \cdot n \cdot M$. We find that: (1) optimal number of rollouts $n$ increases with the compute budget $C$; (2) easy and hard problem sets exhibit similar scaling trends but arise from different underlying mechanisms; (3) under a constraint on $B = B_p \cdot n$, the optimal strategy prioritizes larger $B_p$ (smaller $n$) at low compute budgets, and shifts toward larger $n$ (smaller $B_p$) at high compute budgets to maximize performance; and (4) $B_p$ has only a marginal effect on performance when kept within a moderate range.

problem increases with the sampling compute budget and then saturates. This means that as more compute becomes available, performance improves by allocating more rollouts per problem rather than simply training longer. *Second*, this scaling trend holds across both easy and hard problem sets, but for different reasons. On easy problems, increasing the number of rollouts primarily sharpens performance on already solvable prompts, reflected in improvements in worst@k metrics. On hard problems, larger numbers of rollouts are essential for discovering rare successful trajectories, leading to gains in best@k and improved coverage. *Third*, under fixed hardware constraints (e.g., a fixed number of GPUs), performance is relatively insensitive to the number of unique problems per batch compared to the number of rollouts per problem. This suggests a simple allocation strategy: prioritize sampling more problems when the compute budget admits only a small number of sequential training steps, and shift toward more rollouts per problem as the number of training steps increases. On hard problems, this trade-off is more nuanced and depend on the evaluation metric. *Finally*, while these scaling trends generalize across base models and datasets, the absolute value of the compute-optimal number of rollouts is context-dependent and saturates at different points depending on model capacity, dataset size, and problem difficulty.

## 2. Problem Statement

We consider post-training an LLM using binary outcome-reward RL on a fixed dataset of problems. We focuses on rollout-based on-policy algorithms such as GRPO (Shao et al., 2024), which generate multiple rollouts per prompt and optimize the policy using group-normalized advantages. Concretely, for each prompt, we sample $n$ rollouts, score them with a 0/1 outcome reward, and compute advantages by centering (i.e., subtracting mean) and normalizing (i.e., dividing by standard deviation) rewards *within* this group.

Unlike classical RL, where data acquisition costs arise from interacting with an external simulator, RL for LLMs typically generates its own training data during optimization. As a result, the primary resource constraint is *sampling compute*, which is proportional to the total number of generated rollouts, denoted by $C$. We divide this budget into three parts: **(1)** *problem batch size* ($B_p$), the number of unique prompts sampled per step; and **(2)** *group size* ($n$), the number of parallel rollouts generated per problem in a single update; **(3)** *update iterations* ($M$), the number of sequential gradient updates. $M$ governs the amount of **sequential** compute, while $B_p$ and $n$ govern the amount of **parallel** compute. The effective batch size per iteration is $B = B_p \cdot n$, and the total sampling compute factorizes as $C = B_p \cdot n \cdot M$.

**Formalizing the goal of our study.** Let $\mathcal{A}(B_p, n, M)$ denote an RL algorithm instantiated with these hyperparameters, and let $\mathcal{P}(\cdot)$ denote a scalar performance metric of the resulting model (e.g., reward or pass rate). Our goal is to characterize the scaling laws that govern the optimal allocation of a fixed sampling budget. We formalize this as the following constrained optimization problem:

$$(B_p^*, n^*, M^*) := \arg \max_{B_p, n, M} \mathcal{P}(\mathcal{A}(B_p, n, M))$$

$$\text{subject to} \quad B_p \cdot n \cdot M \leq C_0. \tag{1}$$

This formulation identifies the optimal trade-off between sequential updates and parallel sampling that maximizes performance under a fixed compute budget $C_0$.

## 3. Designing a Healthy RL Recipe

Predictable scaling trends emerge from Equation 1 only if performance $\mathcal{P}(\mathcal{A}(B_p, n, M))$ varies smoothly under compute constraints. A core desideratum, therefore, is that the RL algorithm $\mathcal{A}$ exhibits stable training dynamics as sampling compute is scaled. In practice, naïve RL implementations often violate this requirement (Liu et al., 2025a).

Because hyperparameters such as $(B_p, n, M)$ jointly control both data collection and optimization, changing them *can* induce instabilities in training, making performance highly non-smooth and obscuring underlying scaling structure. Therefore, before studying scaling laws, we first establish a "healthy" RL recipe whose dynamics remain stable across a range of sampling compute budgets. We find that in our setting, training stability is most consistently governed by three factors: (i) problem difficulty relative to the base model, (ii) entropy and KL regularization, and (iii) learning-rate scaling with the effective batch size ($B = B_p \cdot n$).

**Factor 1: Dataset difficulty distribution.** We find that the difficulty of a problem relative to the base model (Snell et al., 2024) strongly affects stability of an RL run. On easy prompts where the base model already samples correct rollouts frequently, RL can quickly drive down entropy and collapse exploration (Cui et al., 2025); on hard prompts, reward is rarely observed and optimization instead demands more exploration. We quantify difficulty by *avg@16*, the base model's average accuracy over 16 rollouts (Qwen2.5-7B-Instruct), which measures the ease of experiencing reward during RL rather than human difficulty. Hence, we construct difficulty-based splits from the Guru-Math dataset (Cheng et al., 2025), each with 300 in-domain validation samples: **(a) Easy**, with $avg@16 \in [0.3, 0.6]$ (6k samples), and **(b) Hard**, with $avg@16 \in [0.0, 0.0625]$ (5k samples). These datasets will be used for our main experiments (Figure 2).

**Factor 2: Entropy and KL-divergence regularization.** Problem difficulty manifests clearly in token-level entropy and, more weakly, in the KL divergence to the base model (Figure 3), both of which serve as sensitive indicators of optimization health. Token-level entropy governs the degree of exploration during generation, while the KL term anchors the policy and limits excessive drift from the base model (Yu et al., 2025). On easy problems, insufficient entropy regularization often leads to premature entropy collapse, causing optimization to stall. In contrast, on hard problems, entropy regularization alone can trigger entropy and response-length explosion, as policy gradients aggressively push toward rare successful trajectories (Qu et al., 2026). In this regime, a KL term can be effective at delaying or preventing early-stage instability, although it is typically unnecessary if training is stable. Hence, whenever we employ an entropy bonus, we pair it with a KL anchor. While applying zero-variance filtering (Khatri et al., 2025) to these terms mitigates instability, we find it suboptimal in performance. In our experiments, we apply both KL and entropy regularization on easy problem sets, where collapse is the dominant failure mode, and remove both on hard problem sets to avoid instability. Importantly, our scaling results are robust to this choice of regularization, provided that training remains stable.

**Factor 3: Learning rate scaling.** Since we vary batch size

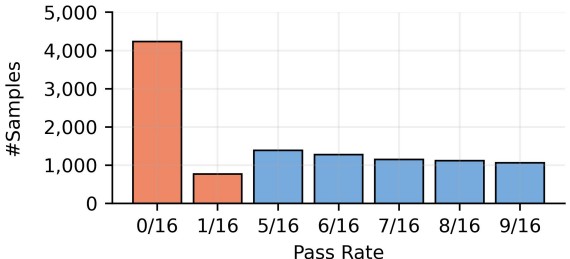

Figure 2. **Difficulty distribution of Easy vs. Hard problems.** We split problems into Easy and Hard sets according to pass@16 (average pass rate over 16 generations per problem).

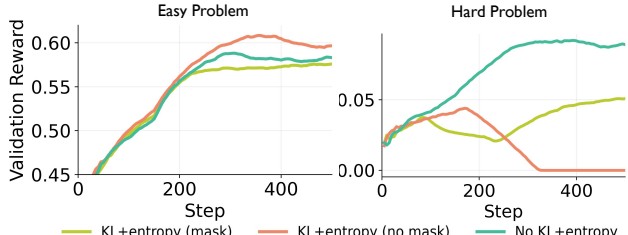

Figure 3. **Regularization ablations on Easy and Hard.** On the Easy set, standard KL+Entropy regularization achieves the best reward. On the Hard set, these regularizers destabilize training even with zero-variance filtering; disabling them yields significantly more stable optimization and higher reward.

($B$) significantly in our scaling laws study, we require a robust LR scaling rule. We first identify a base learning rate $\eta_{base} = 10^{-6}$ at $B = 1,024$ (Figure 4 (left)). Similar to (Yang et al., 2022), we then compare constant, linear, and square-root scaling strategies. As shown in Figure 4 (right), **square-root scaling** ($\eta \propto \sqrt{B}$) provides the best trade-off, enabling faster convergence than constant LR while avoiding the instability of linear scaling.

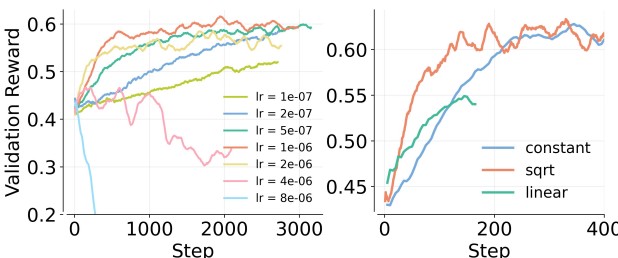

Figure 4. **LR scaling strategy.** Square-root scaling ($\sqrt{B}$) outperforms linear and constant scaling at large batch sizes ($B = 8192$). **Final recipe.** Based on these findings, we adopt the configuration listed in the following for main experiments. See Appendix A for full experiment details in recipe ablations.

| Hyperparameter | Easy | Hard |
|---|---|---|
| KL Regularization | Yes | No |
| Entropy Regularization | Yes | No |
| Zero-var Filter | No | No |
| LR Scaling | $\sqrt{B}$ | $\sqrt{B}$ |

# 4. Allocating Sampling Compute Optimally

We now present empirical results that address our central question: *given a fixed sampling compute budget, how should it be allocated across RL sampling dimensions to maximize performance?* Recall that the total sampling compute scale as $C \propto B_{\mathrm{p}} \cdot n \cdot M$. To study allocation strategies, we sweep over $(B_{\mathrm{p}}, n, M)$ across a range of budgets $C$. For a fixed compute budget $C = C_0$, we evaluate multiple allocations and define the ***compute-optimal frontier*** as the highest i.i.d. validation set reward achievable using total compute $C_0$. Repeating this procedure for increasing values of $C_0$ yields a family of frontiers that characterize how optimal allocation evolves with available compute.

**Data analysis workflow.** To derive our scaling law fits, we subsample each training run to a compact set of ***record-breaking*** points along the learning curve, defined by validation reward as a function of increasing compute. A record-breaking point is the earliest step at which the validation reward exceeds all previously observed values. To robustly identify such improvements, we first discretize validation rewards into bins and then select the earliest point at which the reward enters a higher bin. We then fit a monotonic function to these record-breaking points to obtain prescriptions for the optimal values of $n$, $B_{\mathrm{p}}$, and $M$. Because this preprocessing preserves the ordering of points along the compute axis, it does not introduce spurious non-monotonicity and yields the same performance frontier as fitting over all points (see Appendix A Figure 14 for an illusration).

**Experimental setup.** We sweep over valid configurations $(B_{\mathrm{p}}, n)$, where $B_{\mathrm{p}} \in \{2^5, \ldots, 2^{10}\}$ and $n \in \{2^3, \ldots, 2^{11}\}$, using uniform intervals on a log scale. Due to hardware parallelism limits of the available GPU resources, we additionally impose a hardware constraint $B_{\mathrm{p}} \cdot n \leq B_{\max}$. We set $B_{\max} = 65{,}536$ for the Easy set and $16{,}384$ for the Hard set. The number of sequential update steps $M$ is not fixed and increases as training proceeds. The main experiments used approximately $120{,}000$ H200 GPU-hours. We use a smaller value of $B_{\max}$ for the Hard set to allow for more sequential iterations within a fixed total compute budget. See Appendix A for full experiment setup details. We adopt rollouts rather than tokens as the compute metric, as the number of generated tokens cannot be reliably determined before training and thus provides limited guidance for compute allocation. Appendix E shows that measuring compute in tokens leads to the same log–log scaling behavior.

We study compute-optimal allocation rules under three settings that isolate distinct resource trade-offs: **(1)** $n$ vs. $M$ (parallel rollouts vs. sequential updates); **(2)** $n$ vs. $B_{\mathrm{p}}$ (parallel rollouts vs. number of problems per batch); and **(3)** joint allocation across all resources. Each setting corresponds to a practical scenario in which a practitioner must decide how to allocate limited compute across competing dimensions.

## 4.1. Parallel Samples $n$ vs Sequential Iterations $M$

In this section, we fix the number of problems $B_{\mathrm{p}}$ and study the trade-off between parallel samples $n$ and sequential iterations $M$ under a fixed budget $C$.

**Fitting workflow.** We plot reward vs compute $C$ and fit a ***monotonic sigmoid*** to summarize how the validation set reward (avg@4) scales with compute for that $n$. As mentioned above, we then define the *compute-optimal frontier* as the upper envelope of these fitted curves (see Figure 5). Then, to indicate which $n$ lies on the frontier at each compute level, we color the frontier by $n^*(C)$, which is the value of $n$ whose fitted compute–reward curve achieves the compute-optimal frontier **up to** $C$. Finally, in Figure 6, we fit a log-log plot to show $n^*(C)$ as a function of $C$ to summarize the empirical scaling behavior. We make four important observations in this setting.

*1) The value of $n$ that lies on the compute-optimal frontier shifts systematically higher as the sampling compute $C$ increases (Figure 5).* It is natural to expect larger values of $n$ to be generally favorable at higher compute budgets, analogous to prior work (Hu et al., 2025a), since increasing $n$ lowers policy-gradient variance but it requires more sampling compute. Consistent with this belief, the frontier-attaining $n^*(C)$ shifts to larger values as $C$ grows, and we observe the same trend on both the Easy and Hard problem sets. Smaller values of $n$ exhibit rapid initial gains but plateau at a relatively lower compute regime, whereas larger $n$ sustain improvement over a broader compute range. ***This behavior also suggests that parallel and sequential compute are not exactly interchangeable.*** Choosing $n$ so that we are able to perform a sufficient number of sequential updates $M$ is necessary to achieve strong performance.

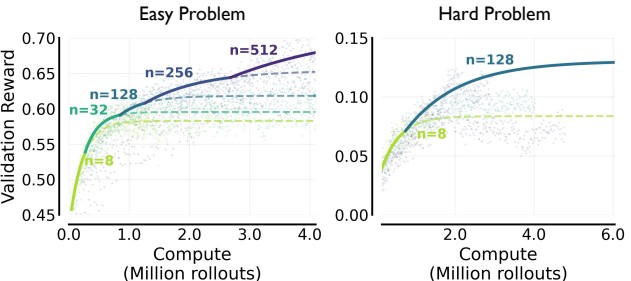

*Figure 5.* **Validation reward frontier vs. compute** ($B_{\mathrm{p}} = 32$). The optimal frontier shifts to larger $n$ as compute increases. For easy problems *(left)*, large $n$ dominates at high compute where small $n$ plateaus. Hard problems *(right)* show the same trend but saturate earlier with a smaller optimal $n$.

*2) Compute-optimal values of $n$ are well-approximated by a sigmoid function of $C$ (Figure 6).* We next aim to fit a functional relationship for the compute optimal value $n^*(C)$ as a function of the available compute $C$. A natural first step is to hypothesize an appropriate functional form. As shown in Figure 6, increasing $C$ admits larger compute optimal

values of $n$, and over a substantial range this relationship appears approximately linear on a log-log scale. The key question is whether this growth continues indefinitely or eventually saturates. Empirically, we observe a clear saturation. Even when evaluating rollout values up to $n = 2,048$, values significantly larger than the saturation point, they fail to extend the frontier, with $n = 512$ continuing to dominate.

We argue that saturation is expected for a fixed base model and a fixed problem set. To build intuition, it is helpful to view increasing $n$ as analogous to spending more compute per gradient step. In supervised learning, increasing capacity alone does not reduce validation error beyond a certain point unless additional training data is available. This principle also underlies pre-training scaling rules from Chinchilla (Hoffmann et al., 2022) that prescribe scaling both pre-training data and model capacity together. Perhaps most closely related to the RL training setup in this study, Setlur et al. (2024) shows that increasing $n$ cannot overcome limitations imposed by a fixed problem set for rejection fine-tuning. As a result, the compute optimal value of $n$ must eventually saturate even for RL, as we observe. We validate this hypothesis regarding a fixed data size in Section 5, where we show how the saturation point shifts given a different base model, problem set size, and distribution.

*3) Next, we find that the compute-optimal allocation trend remains consistent across difficulty levels, although we find harder sets prefer smaller values of $n$ (Figure 6).* We find that the qualitative compute optimal allocation trend remains consistent across problem difficulty. On both easy and hard problem sets, the compute optimal value of $n$ increases with total compute $C$ before eventually plateauing. However, the plateau occurs clearly at smaller values of $n$ on harder problems. In particular, very large values of $n$, such as $n = 512$, yield lower final performance on the hard set and do not lie on the compute optimal frontier. ***This suggests that task difficulty imposes an upper bound on how large $n$ can be used effectively***. While it may seem intuitive that harder problems should benefit from larger $n$ due to increased sampling right away, we observe the opposite behavior in practice. On sufficiently hard problem sets, increasing $n$ allocates substantial compute to problems where the model receives little or no learning signal. In contrast, smaller values of $n$ focus optimization on the subset of prompts where nonzero signal is already present and meaningful improvement is possible. Therefore, it is better to use a smaller value of $n$ to increase the frequency of parameter updates (small $n$, large $M$, more epochs on the same subset of problems) that exploits reachable gains, rather than spending larger $n$ on problems that are persistently unsolved.

*4) Optimization dynamics on the easy and hard sets and the role of various performance metrics (Figure 7).* We saw above that a smaller value of $n$ was more preferable for

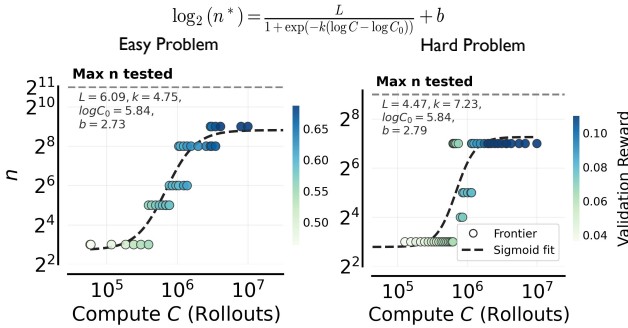

$$\log_2(n^*) = \frac{L}{1 + \exp(-k(\log C - \log C_0))} + b$$

*Figure 6.* **Compute-optimal scaling of the parallel compute $n$** ($B_\mathrm{p} = 32$). The optimal value of rollouts $n$ **shifts systematically higher** as the total sampling compute increases. Points show a running-average estimate of the frontier-attaining $n^*(C)$ at each compute budget (colored by reward), and the red curves fit a sigmoid parameterizing $\log n$ as a function of $\log C$.

optimizing validation *average reward* (avg@4 per problem) and attributed this to solving new problems vs. solving the same problems, but better. We now aim to better understand these optimization dynamics and evaluate how $n^*(C)$ changes if we were to change *the target performance metric we study*. In particular, we consider two metrics: ***best@k*** (or pass@k), defined as the fraction of problems where at least one response out of $k$ is correct, which measures the model's **coverage** over problems; and ***worst@k***, defined as the fraction of problems where all $k$ responses are correct, which we examine to measure the degree to which we can "**sharpen**" around the right solution (i.e., robustness).

Modulo compute-optimality, a larger value of $n$ coupled with as many sequential update steps as needed, should in principle, result in higher values for both *best@k* and *worst@k* on a training dataset. However, this is not quite the case when compute is bounded. We empirically identify the optimal values of $n^*(C)$ for obtaining the highest *best@k* and *worst@k* scores on the validation set, across different $B_\mathrm{p}$ values for the largest value of $C$, and show this number in Figure 7 below. We choose $k = 4 \ll n$ we study, so that none of the trends in Figure 7 are "edge" cases or artifacts of fitting/statistical error. Perhaps surprisingly, we now see an interesting divergence in trends of compute-optimal $n$ that impacts the Easy and Hard sets differently.

**Results.** On the easy set, a larger $n$ is compute-optimal for *worst@4* (sharpening) performance, whereas relatively smaller values of $n$ are compute-optimal for the *best@4* performance. This means that a larger $n$ primarily improves by sharpening more on easy problems, while a smaller $n$ suffices to sample one correct rollout (expected since the set is easy). Conversely, for hard problems, a larger $n$ is more critical for pushing up *best@4* (coverage), while a relatively smaller $n$ is compute-optimal for *worst@4* (sharpening). However, there is a limit beyond which a larger $n$ does not improve coverage on new problems in a compute-optimal way: optimal values here are generally lower than

on the easy set. On the *Extremely Hard* set consisting of all pass@128 = 0 problems (Appendix B; Figure 19), we see a clearer tradeoff of coverage and sharpening: while larger $n$ improves *best@k*, it degrades *worst@k* and lowers average reward. If targeting average reward, the optimal $n$ on hard problems is the value that balances coverage and sharpening well. These results imply that the target metric itself dictates the landscape of compute-optimal $n$.

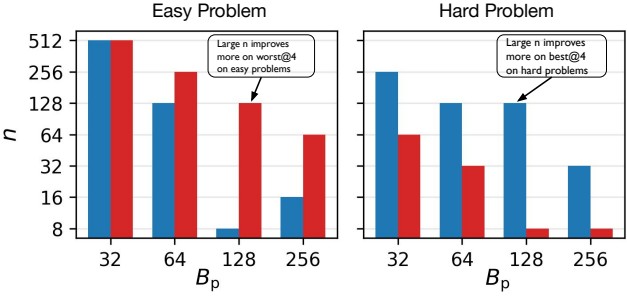

*Figure 7.* **Different mechanisms of how $n$ values optimize *best@4* vs. *worst@4* on easy and hard problems.** Bars show the $n$ maximizing reward for a given $B_p$. On the Easy set *(left)*, the optimal $n$ for best@4 is smaller than for worst@4, indicating that improving robustness requires more parallel rollouts than for coverage. Conversely, on the Hard set *(right)*, a larger $n$ is needed to improve best@4, while worst@4 saturates at smaller $n$.

## 4.2. Bounded Batch Compute: Trading off $B_p$ with $n$

Next, we study a different scaling setup, where we wish to allocate a fixed total batch size $B$ into the number of prompts used and the number of rollouts per prompt used. This question is important in practical settings where hardware parallelism (e.g., number of GPUs or data-parallel) is fixed, and a practitioner needs to make this compute allocation. In such cases, $B$ is often chosen as the largest rollout batch size that saturates sampling throughput ("system batch size"). We additionally experimented with $B_p = 8$ and 16 for the Easy set under fixed $B$ to locate the upper and lower bounds for values of $B_p$ and $n$. We specify the number of sequential iterations $M$ ***a priori*** and seek allocations of $B_p$ and $n$ under a fixed total batch budget $B_p \cdot n \leq B$ that maximize performance. We observe the following:

*1) On the easy problems, allocate more parallel compute $n$ when sequential steps $M$ is large (Figure 8).* In this regime, we examine the compute-optimal value of $n$ under a fixed total batch size (illustrated with $B = 8,192$ only in Figure 8), as $M$ varies. The optimal choice $n^*(M)$ exhibits a sigmoidal dependence on $M$. This behavior suggests that when more sequential updates are available, it is preferable to allocate additional compute toward increasing $n$, rather than increasing $B_p$. The corresponding compute-optimal number of prompts $B_p^*(M)$ decreases with the sampling compute according to an (inverse) sigmoid. In contrast, when $M$ is small, allocating batch size toward a larger $B_p$ is more effective, as it enables many more epochs of training within a given total sequential updates. On the Hard

set, however, the scaling behavior is less consistent. The compute-optimal value $n^*(M)$ exhibits a non-monotonic dependence on $M$ (see Appendix B, Figure 17-18), which implies a similarly irregular trend for the optimal $B_p$. ***This is one of the differences we see across Easy and Hard sets.***

*2) Why do we observe different trends on the Easy and Hard sets in this setup?* As discussed previously, reward can be increased either by scaling $n$, which improves the quality of signal obtained per problem, or by scaling $B_p$, which broadens the set of problems used for training. On the Easy set, where the base model already produces correct rollouts with high probability, the dominant bottleneck is sample quality, making larger values of $n$ preferable as $M$ increases. On the Hard set, however, the optimal allocation depends strongly on the *stage* of training. When the number of sequential updates $M$ is small, low values of $n$ are ineffective at extracting gradient signal, even if training is restricted to a subset of problems. As $M$ increases and the model begins to receive signal on a limited set of problems, increasing $B_p$ becomes preferable, as it prevents overfitting to this small subset. Finally, at larger values of $M$, once training has stabilized across a set of problems, it becomes possible to increase $n$ again without sacrificing coverage, and the compute-optimal allocation shifts back toward larger $n$.

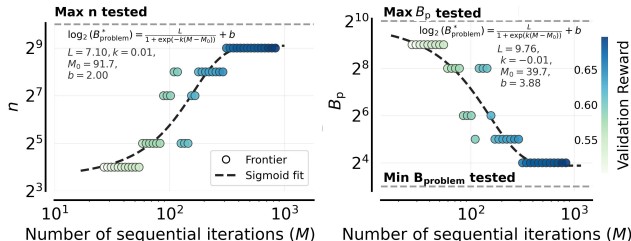

*Figure 8.* **Compute-optimal allocation shifts from $B_p$ to $n$ under fixed total batch size ($B = 8,192 = B_p \cdot n$) on easy problems.** As update iterations ($M$) increases, the optimal allocation follows a sigmoid frontier: $n^*$ increases *(left)* while the optimal number of prompts $B_p^*$ inversely decreases *(right)*, indicating a shift toward higher per-problem sampling at larger budgets.

To make the above argument concrete, we study the effect of varying $B_p$ at fixed $n$, as well as varying $n$ at fixed $B_p$, and assess which hyperparameter more strongly influences validation performance. On the Easy set, we find that changing $B_p$ has only a marginal effect on validation reward, whereas increasing $n$ leads to substantial gains up to saturation (Figure 9, left). This explains the sigmoidal scaling behavior observed earlier: since performance is primarily driven by $n$, increasing $n$ is preferred at larger compute budgets, with $B_p$ decreasing accordingly under a fixed batch size constraint. On the Hard set, the picture is more nuanced (Figure 9, right). While increasing $n$ remains beneficial, varying $B_p$ produces performance changes of comparable magnitude, and overall sensitivity to both hyperparameters is weaker. As a result, the compute-optimal choice of $n$ is noisier, and at intermediate values of $M$, allocating additional compute

toward increasing $B_p$ can yield better performance.

**To put these findings together,** we find that setting a large $n$ (up to the saturation point), with a moderate $B_p$, is the most robust strategy. For example, in our experiments, we observed no significant threshold effects for $B_p$ between 32 and 1024 on the Easy set. However, on the Hard set or a skewed problem distribution, we speculate they may require a higher minimum $B_p$ for effective training.

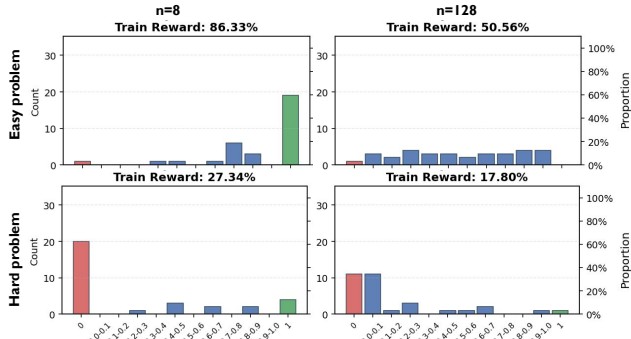

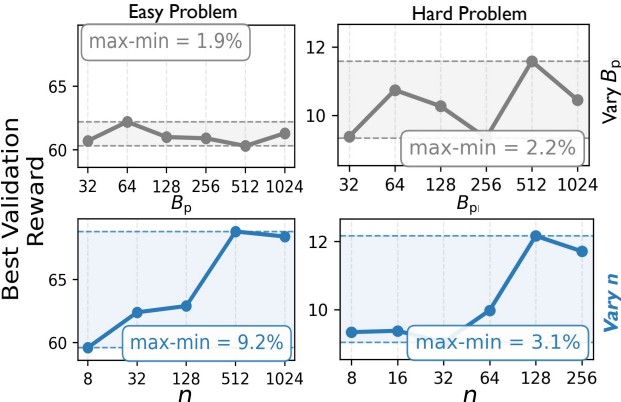

*Figure 9.* **Sensitivity of validation reward to $B_p$ vs. $n$.** Easy *(left)*: The impact of varying $n$ (9.2% range) shows a clear positive correlation and is significantly larger than varying $B_p$ (1.9%). Hard *(right)*: Sensitivity to $B_p$ (2.2%) becomes comparable to $n$ (3.1%). The fluctuating trend in the top-right plot suggests that $B_p$ selection introduces optimization instability on hard tasks, explaining the less predictable allocation trends when fixing $B$.

### 4.3. Jointly optimizing $(B_p, n, M)$

Finally, we relax all constraints and jointly optimize the three sampling axes $(B_p, n, M)$ under a fixed total rollout compute budget $C = B_p \cdot n \cdot M$. The compute-optimal solution is still largely governed by $n$: *the optimal $n^*(C)$ follows the same sigmoidal scaling with compute (Figure 21)*. In contrast, $B_p$ mainly serves as a stability knob and has only a marginal impact on performance within a moderate range. Practically, we tune $n$ via $n^*(C)$, pick the smallest stable $B_p$, and assign the remaining budget to $M$. Joint frontiers and sigmoid curves are in Appendix D. We also show scaling $n$ improves not only in-domain validation, but also OOD downstream tasks in Appendix C (Figure 20).

## 5. Role of Base Model and Prompt Set

Having seen that the compute-optimal number of rollouts $n$ increases with sampling compute $C$ on both Easy and Hard sets, it is natural to ask whether this behavior extends to other prompt distributions and base models. We now address this question, with the goal of identifying the factors that govern scaling trends underlying our findings.

*Figure 10.* **Training reward distributions on Easy and Hard sets at the same compute ($n = 8$ vs. $n = 128$).** (1) Interference: On the Easy set (initial pass rates 0.3–0.6), optimization sacrifices some problems, leaving a non-zero fraction unsolved after training. (2) Easy set: Larger $n$ results in a more uniform distribution of pass rates, avoiding polarized outcomes seen in smaller $n$. (3) Hard set: Larger $n$ improves coverage (reducing zero fraction), while smaller $n$ sharpens performance on a subset of solvable problems.

### 5.1. Addressing Interference Requires Scaling $n$

As we show in Appendix G, if we were given a multi-armed bandit problem, in a tabular setting, the compute-optimal scaling law would prescribe increasing $M$ (sequential updates) over $n$. However, this prediction contradicts our empirical findings that show scaling $n$ is better. In this section, we argue that this gap arises due to *interference* across problems (Schaul et al., 2019). When multiple problems are trained jointly, gradient updates can interfere, possibly causing uneven learning across problems and degradation on previously solvable problems. In this regime, allocating compute toward larger $n$ is preferable to increasing $M$, since more rollouts yield more uniform updates across problems per step and improve learning efficiency. This shifts the compute-optimal balance toward parallel sampling rather than sequential optimization, mitigating interference and improving learning efficiency.

**Evaluating interference.** To quantify interference, we analyze the training-set pass@1 distribution across problems under matched compute budgets ($n \cdot M$). Even on the Easy set, a non-trivial fraction of problems end training with pass@1 close to zero, indicating uneven progress across problems. Under the same compute budget, larger values of $n$ yield a less skewed distribution and more uniform improvements (Figure 10). A similar pattern holds on the Hard set: smaller $n$ optimizes on a subset of problems while leaving many unsolved, whereas larger $n$ reduces the zero-pass fraction and improves coverage. Overall, increasing $n$ mitigates interference by distributing updates more evenly across problems, explaining why larger $n$ is preferred.

**Compute-optimal $n$ scaling generalizes for different base models.** As shown in Figure 11, larger $n$ values consistently outperform the baseline ($n = 8$) at high compute budgets for both Qwen3-4B-Instruct and Llama 3.1-8B-Instruct on their

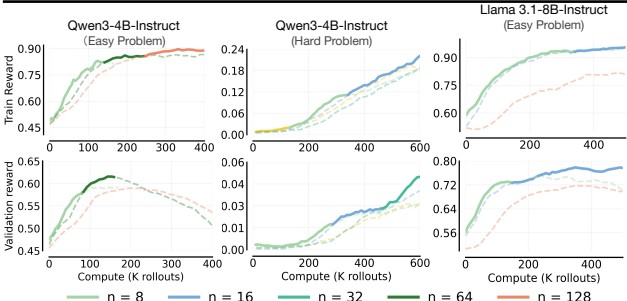

*Figure 11.* **Generalizing $n$ scaling trends to other models.** We observe increasing $n$ boosts returns at high compute across all settings, while optimal $n$ saturates differently.

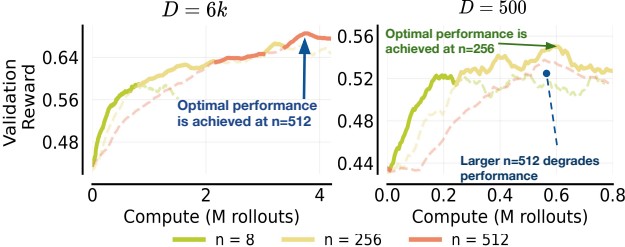

*Figure 12.* **Impact of data size ($D$) on frontiers.** With more data ($D = 6k$; ***left***), performance scales up to $n = 512$. With small data ($D = 500$; ***right***), the frontier saturates at smaller $n = 256$, and scaling further to $n = 512$ leads to overfitting and degradation.

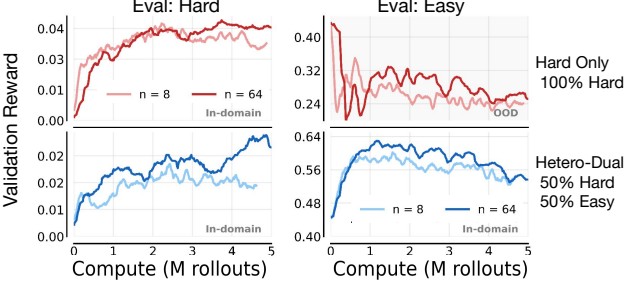

*Figure 13.* Different data distributions (5K samples) yield results with $n \in \{8, 64\}$. Row 1: Hard Only; Row 2: Dual Mix (50% Hard/Easy). **We see larger $n$ improves performance on both extremely hard and mixed distributions**. More findings: (1) Hard-only training causes catastrophic forgetting on easier tasks; (2) Mixing easy data mitigates this with loss on hard tasks.

corresponding Easy and Hard sets. However, the *values* of $n$ are specifically model-dependent, and largely depend on the interaction between the base model and the problem set. We also observed that the validation reward for both models ceases to increase or degrades at $n = 128$ on both Qwen3 and Llama3.1 on easy problems. This saturation in validation occurs while the ***training reward continues to rise***. We attribute this divergence to the train-test gap (overfitting), discussed next.

### 5.2. Train-Test Gap

Our scaling results use validation metrics, even though optimization dynamics are primarily driven by the training set. Therefore, scaling laws on the validation set require sustained transfer from training to test, which is unguaran-

teed. When the prompt set is too small, training may overfit prematurely. In such cases, larger $n$ may no longer appear compute-optimal, even at high compute budgets, because additional training fails to improve validation performance. Figure 12 shows that when we vary the prompt set size $D$, that compute-optimal $n$ caps out at smaller values when $D$ is smaller. This is expected: validation reward degrades under prolonged training due to overfitting, preventing larger $n$ from attaining the frontier. Therefore, we find that optimal allocation for training performance may not agree with that of the validation set, especially at large compute budgets.

### 5.3. Dataset Skew: Other Data Compositions

Finally, we train on heterogeneous mixtures of Easy and Hard problems (Figure 13), as well as an "extra hard" set where the base model attains pass@128 = 0. These mixtures induce different skewness and thereby alter the rate at which pass@1 improves in training. Despite this variation, we observe a consistent crossover trend that larger $n$ outperforms smaller $n$ on validation sets. The compute ranges where small $n$ is optimal are different. This suggests the rate of pass@1 improvement controls both the compute range over which a given $n$ is optimal and the minimum compute-optimal $n$. Crucially, our central finding remains unchanged: ***larger compute budgets $C$ support larger compute-optimal values of $n$, even under skewed mixtures***.

## 6. Related Works

Scaling laws are well established for pretraining (Hestness et al., 2017; Kaplan et al., 2020; Hoffmann et al., 2022), but predicting RL behaviors is more challenging due to coupled data collection and optimization. Prior work reports approximate power-law scaling in controlled RL settings such as board games and single-agent deep RL (Jones, 2021; Hilton et al., 2023), and characterizes compute–data trade-offs and Pareto frontiers in value-based RL (Fu et al., 2025; Rybkin et al., 2025). Whether such predictability extends to LLM RL remains unclear, as experience is generated on-policy at high cost and scaling behavior depends on recipe-level stability. Recent studies make progress by extending on-policy RL under fixed pipelines and observing sigmoidal reward–compute curves (Khatri et al., 2025), or varying model size (Tan et al., 2025). However, instabilities such as entropy collapse or policy drift, often require stabilizers including KL, clipping, or resets (Cui et al., 2025; Hu et al., 2025a). As a result, existing work largely *describes* scaling along fixed recipes, whereas practitioners face a *budgeted* problem: how to allocate a fixed sampling budget across the degrees of freedom that jointly govern exploration and learning in on-policy LLM RL. We therefore study RL scaling laws as *prescriptive allocation rules*, using compute-optimal analysis over $(B_p, n, M)$ under stable recipes. See Appendix I for related works on scaling specific dimensions.

# 7. Conclusion

We studied the optimal compute allocation in LLM RL across update iterations ($M$), problem batch size ($B_p$), and group size ($n$). We found optimal $n$ follows a sigmoidal scaling law with compute and the underlying mechanisms depend on difficulty: easy problems favor sharpening, while hard ones necessitate broader coverage. We provide an extended discussion on future perspectives in Appendix J.

## Acknowledgements

We thank Oleg Rybkin, Apurva Gandhi, Charlie Snell, Matthew Yang, Rishabh Agarwal, Sang Michael Xie, Junlong Li, Zora Wang, and other members of the CMU AIRe lab for their thoughtful feedback and discussions. We also thank Chengyu Dong, Mikhail Yurochkin, Rupesh Srivastava, Joel Hestness, and Gavia Gray for early discussions on RL scaling in LLM. We also gratefully acknowledge the Orchard cluster at the FLAME center of CMU for providing computational resources that supported a part of this work.

## Impact Statement

This work studies how to allocate sampling compute more efficiently for reinforcement learning (RL) post-training of large language models. By providing prescriptive guidance on the choice of parallel rollouts, batch composition, and sequential updates under fixed compute budgets, it aims to improve the efficiency, predictability, and reproducibility of RL research workflows rather than to introduce new model capabilities.

More efficient RL post-training may indirectly accelerate the development of models with stronger reasoning or problem-solving abilities. As with most advances in training efficiency, this could amplify both beneficial and harmful downstream uses of language models. However, this work does not introduce new objectives, reward functions, or data sources, nor does it propose methods that bypass existing safety or alignment mechanisms. Instead, it analyzes the behavior of standard on-policy RL methods under different compute allocations, with recommendations that remain bounded by the capabilities and limitations of the underlying base models and datasets.

A primary societal benefit of this work is its potential to lower barriers to RL experimentation with large models. RL post-training is often computationally expensive and difficult to tune, making it accessible mainly to well-resourced institutions. By clarifying which scaling strategies are compute-optimal and which exhibit diminishing returns, our results may help smaller research groups use limited resources more effectively, reducing wasted compute and associated environmental costs. In this sense, this work contributes to more sustainable and democratized research

practices.

Overall, this work is best understood as a methodological contribution that improves how RL experiments are designed and interpreted, with broader impacts arising primarily from increased efficiency, accessibility, and rigor in LLM RL post-training research.

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

# A. Detailed Experiment Setup

**Recipe ablation setup.** We use Qwen2.5-7B-Instruct (max length 8,192) with GRPO. For regularizer ablations, we fix $B_p = 256$ and $n = 16$. On both Easy and Hard sets, we ablate KL and entropy regularization and the zero-variance filter (including applying it selectively to loss terms). For LR scaling, we use AdamW (Loshchilov & Hutter) with a 10-step linear warmup followed by a constant schedule. We establish a base LR anchor at $B_p = 128, n = 8$ ($B = 1,024$) via grid search. We then scale to $n = 64$ ($B = 8,192$) to compare three scaling rules: (1) constant, (2) linear, and (3) square-root scaling.

**Zero-variance filtering** is employed in recent works (Khatri et al., 2025) to exclude prompts with identical rollout rewards from loss in GRPO. This mechanism increases effective batch size and prevents applying regularizers to zero-gradient trajectories, a crucial feature for hard problems where exploration naturally drives high entropy. However, our experiments (Figure 3) show that even when filtering is *applied to KL and entropy terms*, instability and entropy explosion persist, though mitigated, when rare positives are sampled. Since removing regularization entirely yields the most stable dynamics, we employ KL+entropy regularization only on the Easy set and omit them on the Hard set to avoid instability.

**Main experiment setup.** We train Qwen2.5-7B-Instruct with on-policy updates using the optimized recipe above. The learning rate scales proportionally to $\sqrt{B}$ (base 1e-6 at $B = 1024$). Based on ablation results, KL and entropy regularization are enabled for the Easy set but disabled for the Hard set. We fix temperature to 0.6 and top-$p$ to 1.0. We use the GRPO algorithm and Truncated Importance Sampling (TIS (Yao et al., 2025)) to mitigate training-inference logit mismatch. We use the veRL (Sheng et al., 2024) framework to conduct all RL experiments.

**Extracting frontiers.** Figure 14 provides a schematic illustration of how we extract frontiers and fit the sigmoidal curve.

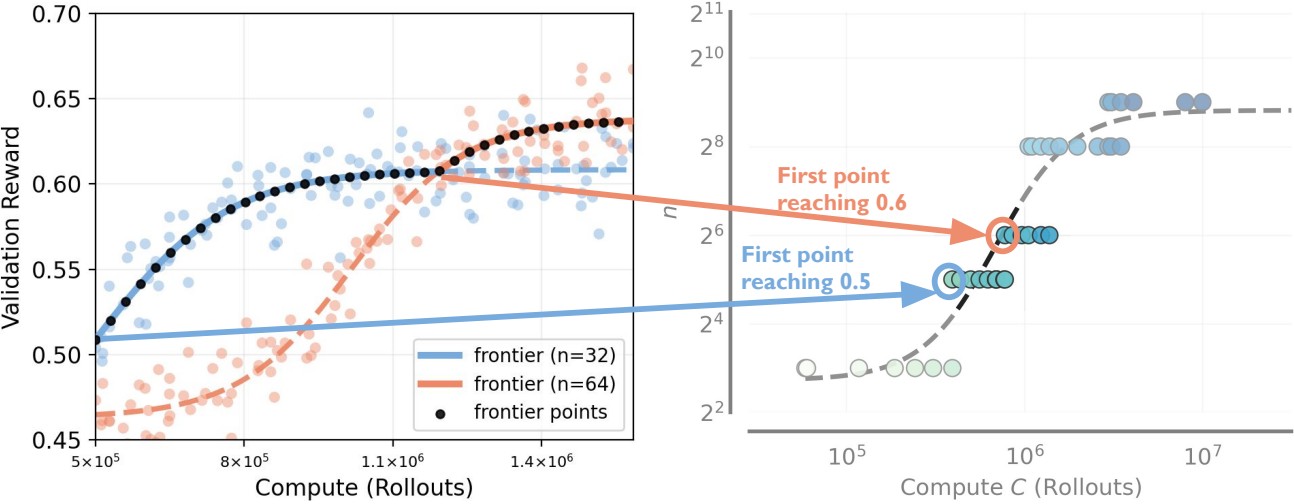

*Figure 14.* **Demonstrations of frontier point detection for each** $n$. **(Left)** Validation reward trajectories plotted against compute (rollouts) for varying population sizes ($n = 32$ in blue, $n = 64$ in red). Scatter points show raw data; dashed curves show smoothed trends. Arrows illustrate the "record-breaking" extraction process, identifying the earliest compute step where reward crosses a discretized threshold (e.g., 0.5 or 0.6). In practice, we employ finer reward bins (e.g., 0.005) tailored to task difficulty. **(Right)** Extracted frontier points in the $n$ vs. Compute space. Each circle represents the compute budget $C$ required for a specific $n$ to reach a higher performance bin. The dashed curve shows the fitted scaling law, indicating the optimal $n$ scaling as compute increases.

# B. Additional Compute-Optimal Results

In the main results, we show one fixed value for $B_p = 32$ for brevity. Figures 15 and 16 demonstrate that the scaling trend described in the main text, where larger compute budgets favor increased parallel rollouts ($n$), holds across different fixed values of $B_p$. While it appears that larger $B_p$ settings saturate at lower $n$ values (e.g., $n = 16$ at $B_p = 1,024$), this might be attributable to the total batch size constraint ($B_{max} \geq B_p \cdot n$) in the sweep experiments. The precise interaction between $B_p$ and the saturation point of $n$ remains an open question for future investigation.

Figure 17 and 18 provide additional compute-optimal frontiers under different fixed values of $B_p$ on the Easy and Hard splits. Consistent with Section 3.2, higher sampling budgets increasingly favor larger $n$, indicating that allocating more parallel rollouts per problem is a robust strategy across dataset difficulty and batch-size settings.

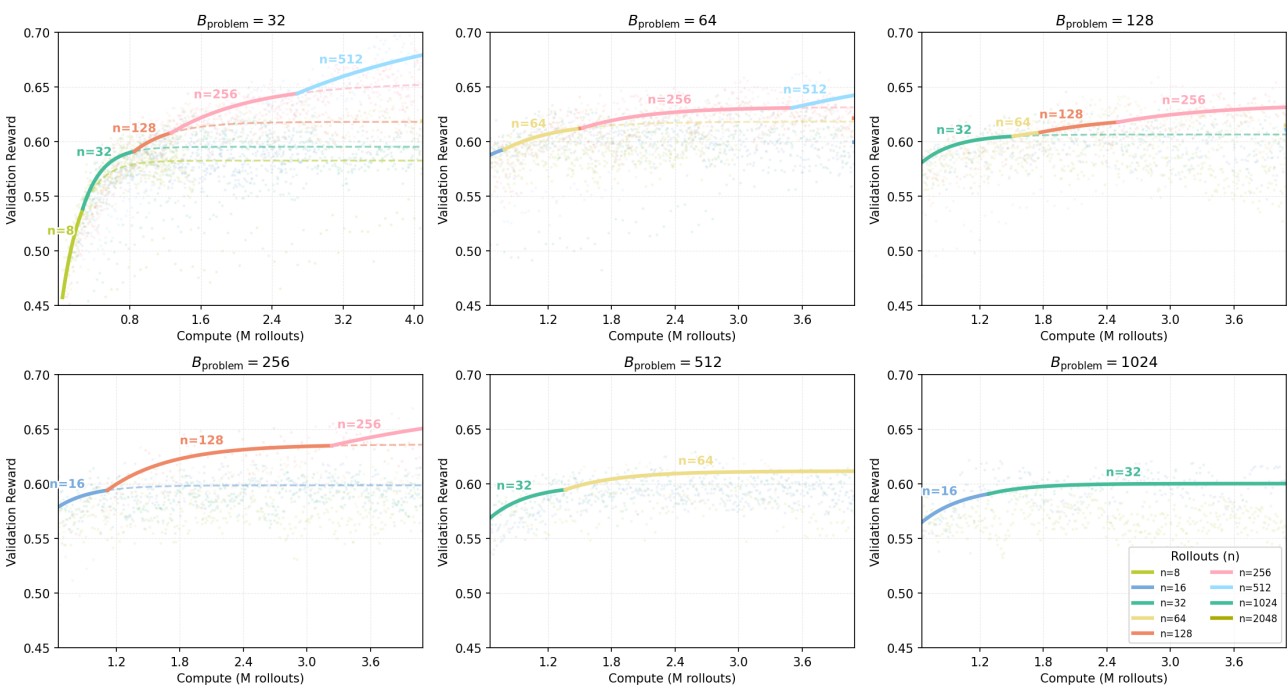

*Figure 15.* **Compute-optimal frontiers maximizing over $n$ varying problems per batch ($B_\mathbf{p}$) on the Easy set.**

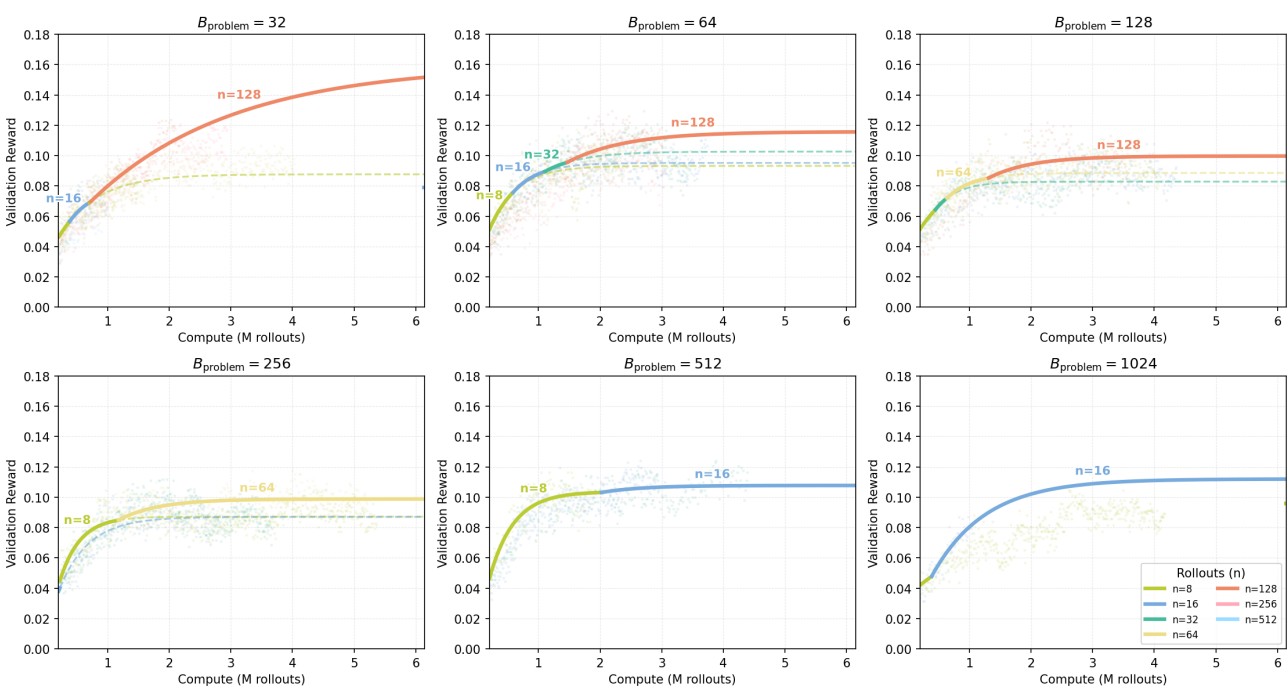

*Figure 16.* **Compute-optimal frontiers maximizing over $n$ varying problems per batch ($B_\mathbf{p}$) on the Hard set.**

Finally, we report results on the in-domain *Extremely Hard* subset (pass@128 = 0) using both **best@4** and **worst@4** metrics (Figure 19). We observe a clear **coverage–sharpening trade-off**: larger $n$ is more beneficial for improving **best@4** (coverage), while **worst@4** (sharpening) is compute-optimally maximized at a **moderate** $n$ (e.g., $n = 64$). Notably, overly large $n$ (e.g., $n = 256$) can underperform on worst@4 despite achieving better coverage, suggesting that the compute-optimal choice of $n$ on extremely hard problems typically lies in an intermediate regime that balances exploration and consistency.

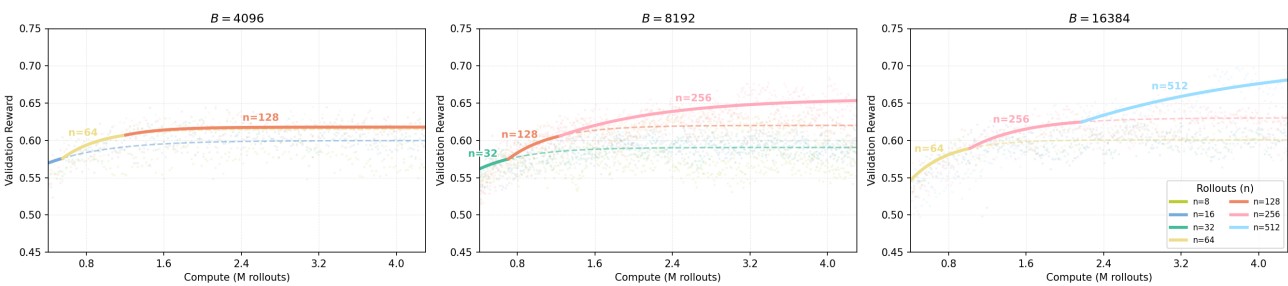

*Figure 17.* **Compute-optimal frontiers on the Easy set under fixed total batch size** $B \in \{4096,\ 8192,\ 16384\}$. Each subplot fixes the total batch size $B$ and sweeps the number of parallel rollouts per problem plotting validation reward versus compute (measured in millions of rollouts).

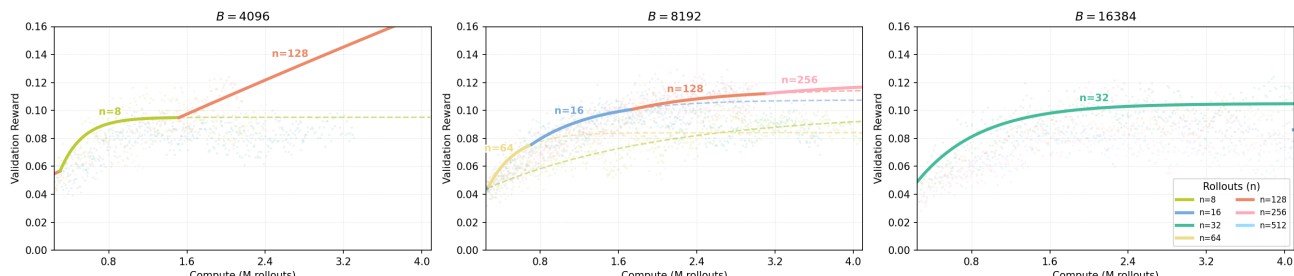

*Figure 18.* **Compute-optimal frontiers on the Hard set under fixed total batch size** $B \in \{4096,\ 8192,\ 16384\}$. Compared to the Easy set, the trends are **noisier** in the Hard regime. Nevertheless, the qualitative trend remains consistent: as compute increases, the compute-optimal allocation increasingly favors larger parallel rollouts per problem, i.e., larger $n$.

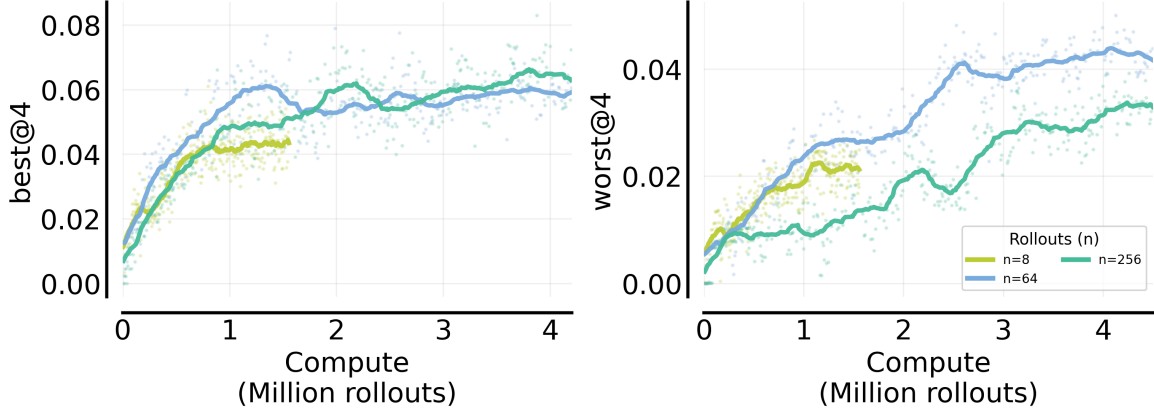

*Figure 19.* **Compute-optimal frontiers on the in-domain Extremely Hard subset (pass@128 $= 0$), evaluated with best@4 (left) and worst@4 (right).** Larger $n$ improves **best@4** at higher compute, whereas **worst@4** is maximized by a **moderate** $n = 64$, highlighting a strong coverage–sharpening trade-off in the extremely hard regime.

## C. Generalization Experiments

In the main text, we prioritize in-domain validation results to minimize the influence of train-test distribution shifts, thereby allowing for a cleaner analysis of compute allocation scaling. In reality, practical post-training workflows require models to generalize to unseen distributions and downstream tasks. We therefore examine whether the benefits of increasing parallel rollouts ($n$) extend beyond the main Guru-Math setting.

## C.1. Out-of-domain (OOD) Math Evaluation

Beyond the in-domain math evaluations, we evaluate whether the same trend transfers to AIME 2024, an out-of-domain math benchmark. As illustrated in Figure 20, we observe that larger values of $n$ lead to higher performance on AIME 2024.

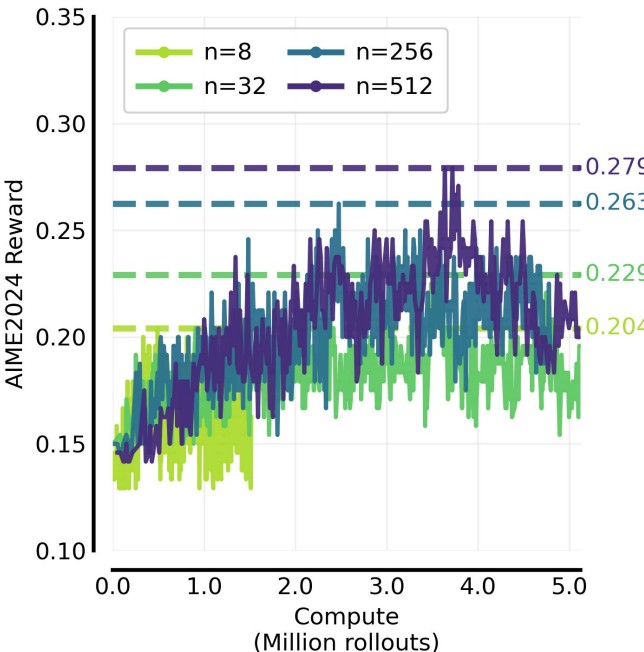

*Figure 20.* AIME 2024 scores of models trained with varying parallel rollouts ($n$) under a fixed problem batch size ($B_p = 32$).

## C.2. Generalization Across Model Scales and Domains

We further evaluate whether the same qualitative trend holds beyond the original model scales and math-only setting. Specifically, we run additional experiments on larger models, including Qwen3-32B and K2-V2 70B (Team, 2025; Liu et al., 2025b), and on new domains including algorithmic coding, logic puzzles, and a long-horizon SWE-agent environment. For Code and Logic, we use the domains from Guru (Cheng et al., 2025), sampling 1000 training examples and evaluating on four held-out splits of 200 examples each. For the SWE-agent setting, we evaluate on R2E-Gym (Jain et al., 2025) using Qwen3-32B, with a maximum of 50 turns per episode and 64k-token maximum length. For K2-V2 70B, we train on Guru-Math with 64k-token maximum length.

*Table 1.* Generalization experiments across model scales and domains. Each entry reports the best reward achieved, with the compute required to reach it shown in parentheses.

| Model | Task | $n = 8$ | $n = 64$ | $n = 128$ |
|---|---|---|---|---|
| Qwen2.5-7B-Instruct | Guru-Code | 68.7 (0.2M) | 69.8 (0.4M) | **73.9** (0.5M) |
| Qwen2.5-7B-Instruct | Guru-Logic | 30.7 (770K) | **31.9** (2.7M) | 31.3 (4M) |
| Qwen3-32B | SWE agent | 35.9 (118K) | **43.5** (147K) | – |
| K2-V2 70B | Guru-Math | 85.0 (389K) | **87.1** (1.3M) | – |

Across these settings, we observe the same qualitative pattern as in the main paper: larger values of $n$ tend to become preferable at higher compute budgets. The sparse sweeps are also consistent with the saturation behavior observed in the main experiments, although they are not intended to precisely identify the saturation point in each new setting. These results suggest that the main phenomenon is not confined to the original math setting or to smaller model scales, although the exact saturation point can vary across models, domains, and compute regimes.

## C.3. Generalization Across RL Algorithms

We also test whether the trend holds beyond GRPO by experimenting with PPO and CISPO (Schulman et al., 2017; Chen et al., 2025). The training and validation data are Guru-Math, and the base model is Qwen2.5-7B-Instruct.

*Table 2.* Generalization experiments across RL algorithms on Guru-Math with Qwen2.5-7B-Instruct. Each entry reports the best reward achieved, with the compute required to reach it shown in parentheses.

| Algorithm | Data | $n = 32$ | $n = 64$ | $n = 128$ | $n = 256$ |
|---|---|---|---|---|---|
| PPO | Easy | 58.8 (2.6M) | 62.3 (2.3M) | **64.8** (2.7M) | 64.3 (2.8M) |
| PPO | Hard | 15.7 (1.1M) | 17.1 (2.2M) | **17.9** (1.8M) | 17.0 (2.1M) |
| CISPO | Easy | 61.3 (3.1M) | – | – | **69.4** (4.6M) |

The PPO and CISPO results are consistent with a similar qualitative trend: larger $n$ generally becomes more favorable as compute increases, and the best-performing configuration often lies at a larger rollout count. This suggests that the compute-allocation pattern is not specific to GRPO. At the same time, because these experiments are sparser than the main sweeps, we treat them as supporting evidence for generality rather than as a separate scaling-law fit.

# D. Additional Details: Joint Optimization of $(B_{\mathbf{p}}, n, M)$

In Section 4.3, we jointly optimize the three sampling axes $(B_{\mathbf{p}}, n, M)$ under a fixed total rollout compute budget

$$C = n \cdot B_{\mathbf{p}} \cdot M. \tag{2}$$

For each compute budget $C$, we exhaustively sweep a grid of feasible pairs $(B_{\mathbf{p}}, n)$ within the range accessible to our system, and set

$$M = \left\lfloor \frac{C}{n\, B_{\mathbf{p}}} \right\rfloor \tag{3}$$

(up to standard feasibility constraints such as minimum required update steps and hardware throughput limits). We then select the best configuration at each $C$ by

$$(B_{\mathbf{p}}^*(C), n^*(C), M^*(C)) = \arg \max_{(B_{\mathbf{p}}, n, M) \in \mathcal{G}(C)} \mathrm{Reward}_{\mathrm{val}}(B_{\mathbf{p}}, n, M), \tag{4}$$

where $\mathcal{G}(C)$ denotes the feasible sweep grid at budget $C$ and the validation metric is avg@4 unless stated otherwise.

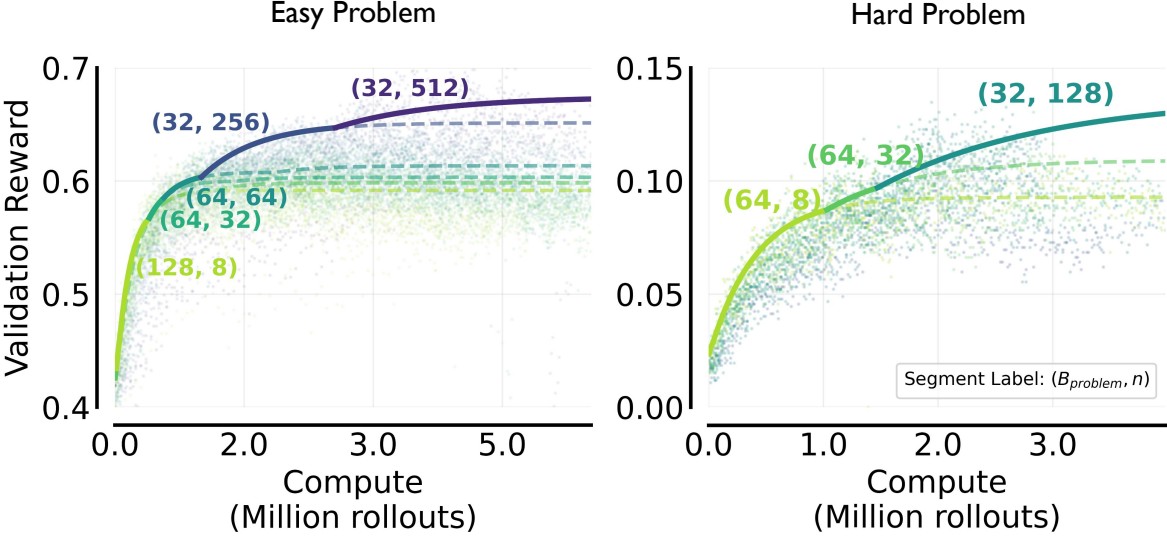

*Figure 21.* **Compute-optimal parallel rollouts $n^*(C)$ under joint optimization of $(B_{\mathbf{p}}, n, M)$.** For each total rollout compute budget $C$, we sweep $(B_{\mathbf{p}}, n, M)$ and select the globally best configuration. The optimal $n$ increases monotonically with compute and is well-fit by a sigmoid trend on both the Easy **(left)** and Hard **(right)** splits.

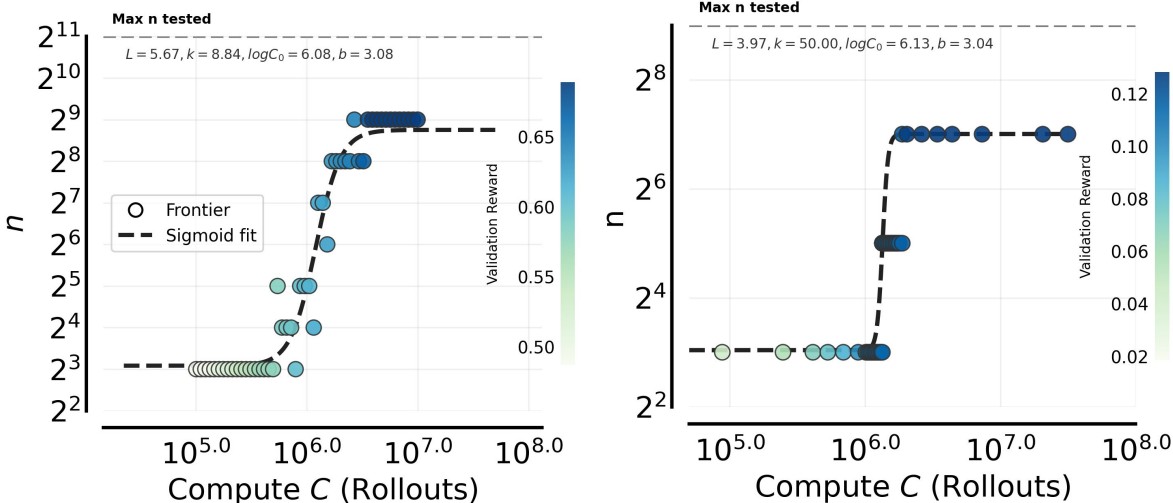

*Figure 22.* **Compute-optimal frontiers from sweeping** $(B_\text{p}, n, M)$ **on Easy and Hard problems.** Points on the frontier are annotated by the pre-training sampling configuration $(B_\text{p}, n)$, with $M$ determined by the remaining compute. Consistent with earlier sections, the frontier shifts to systematically larger $n$ as compute increases. In contrast, the frontier-attaining $B_\text{p}$ varies across budgets but has only a marginal effect on performance within a moderate range (cf. Section 4.2).

Across both easy and hard splits, the joint sweep confirms a consistent pattern: the compute-optimal strategy is primarily characterized by the parallel rollouts per problem. As shown in Fig. 21–22, $n^*(C)$ increases monotonically with compute and is well-fit by a sigmoid trend in $\log n$ versus $\log C$. In contrast, $B_\text{p}$ behaves mainly as a *stability constraint* rather than a performance driver: once $B_\text{p}$ is kept within a moderate range, performance varies only weakly with $B_\text{p}$, and multiple $B_\text{p}$ values can yield similarss results provided training remains stable. In practice, we therefore recommend the following workflow: (i) tune $n$ using the fitted $n^*(C)$ curve, (ii) choose the smallest $B_\text{p}$ that yields stable training for the target difficulty regime, and (iii) allocate the remaining budget to $M$.

Finally, we note that while our sweeps are exhaustive over the $(B_\text{p}, n)$ range we could access, we do not explore regimes with extremely large total rollout sizes where both $B_\text{p}$ and $n$ are simultaneously large; understanding interactions at such massive batch sizes is an important direction for future work.

## E. Compute Metrics: Rollouts vs. Tokens

To verify that our compute–optimal $n^*$ scaling is not an artifact of how we measure compute, we repeat the same fit using another unit: **total generated tokens**. As shown in Figure 23, both parameterizations lead to an almost identical sigmoid trend. This suggests that, for our training setup, using rollouts or tokens as the compute proxy makes little practical difference. The two views are largely related by a near-constant conversion factor governed by the average response length.

One noticeable difference is that the fitted slope parameter $k$ is not exactly the same across the two plots. This is expected: $k$ controls how sharply $n^*$ transitions as compute increases, and its numerical value depends on the units of $C$. In experiments, we observe a positive correlation between the model's response length and validation rewards. For instance, models at the high-compute frontier tend to have longer response lengths. Since token-based compute accounts for response length, the $k$ value is smaller, indicating a shallower slope in $n$ scaling relative to compute. Therefore, the change in $k$ mainly reflects how response length modulates the mapping between rollouts and tokens, rather than a fundamental discrepancy in the underlying scaling behavior. Nonetheless, the overall scaling trend remains consistent.

## F. Effects of Reducing Baseline Estimation Variance

We discuss in the main content how larger $n$ outperforms small $n$ at high compute regimes from exploration and optimization perspectives. Another theoretical advantage of larger $n$ in GRPO is providing a more robust baseline estimator (group average reward), thereby reducing advantage estimate variance. To isolate the gain attributed specifically to precise baseline estimation versus training on more data, we conducted an ablation with a fixed problem batch size ($B_\text{p} = 128$). We compared

$$\log_2\left(n^*\right) = \frac{L}{1 + \exp(-k(\log C - \log C_0))} + b$$

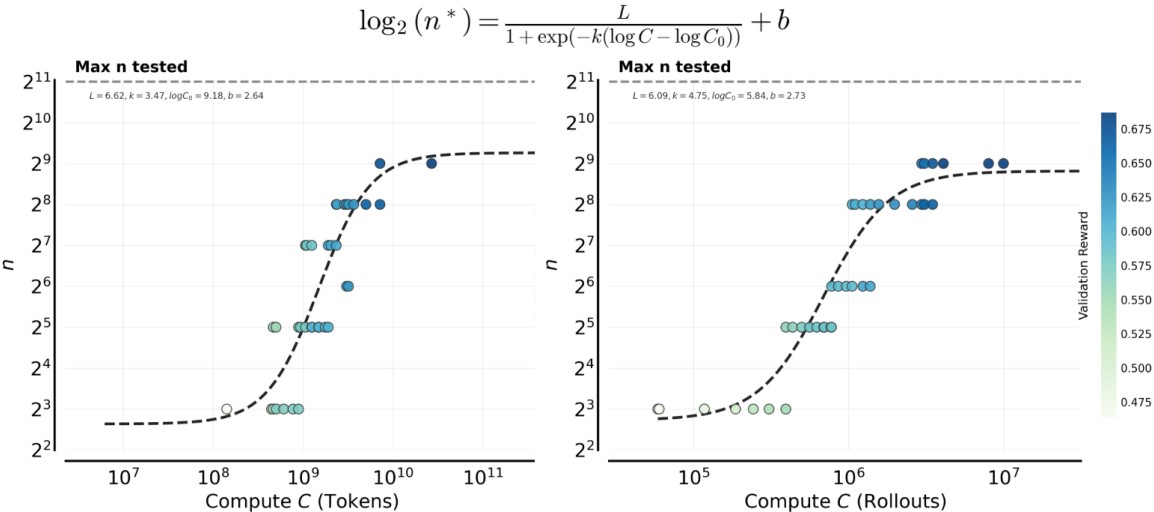

*Figure 23.* $n^*$ **scaling is consistent under token-based vs. rollout-based compute.** We fit sigmoid curves for $\log_2(n^*)$ as a function of compute $C$, using either total generated tokens (**left**) or total rollouts (**right**). Both choices produce the same qualitative scaling curve—rapid growth followed by saturation—indicating that the compute-optimal $n^*$ trend is robust to the compute definition.

three settings: **(1) Large** $n = 256$, **(2) Small** $n = 64$, and **(3) Decoupled**, where we generate 256 rollouts to compute high-precision advantage estimates but randomly subsample only 64 rollouts for the policy gradient update.

We observe the best validation reward follows **(1) > (3) ≈ (2)**. The fact that (3) performs similarly to (2) indicates that the benefit of a lower-variance baseline estimator is not significant in this context. Consequently, the superior performance of (1) over (3) suggests that the primary benefit of scaling $n$ stems from broader exploration rather than baseline precision.

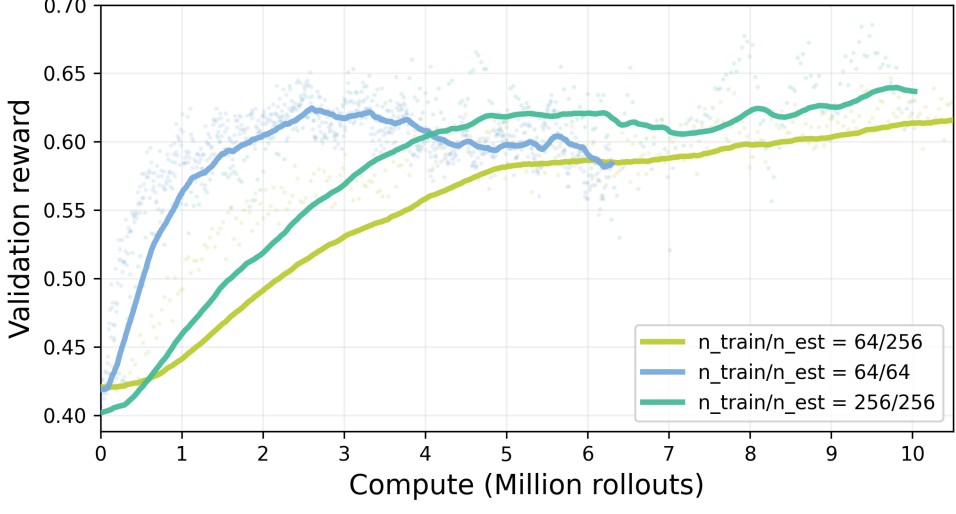

*Figure 24.* **Effects of baseline estimation variance.** Validation reward vs. compute (million rollouts) under a fixed problem batch size $B_p = 128$, comparing three GRPO settings: (i) large group size $n_{\text{train}}/n_{\text{est}} = 256/256$, (ii) small group size $64/64$, and (iii) decoupled baseline estimation $64/256$ (estimate baseline from 256 rollouts but sample 64 from them for the policy-gradient update). We observe consistent ordering **(1) > (3) ≈ (2)**, showing that lower-variance baseline estimation yields negligible gains, while the full $n = 256$ run remains best, indicating the dominant gains from scaling $n$ come from broader exploration.

# G. Base Case: Only One Training Problem

To build a conceptual model, let us study the simplest setting where we are provided with *one single problem* in the training set. We model this setting as a simple multi-armed bandit problem, where each arm represents one possible response to the

problem. We assume training of a tabular softmax policy (i.e., softmax on independently represented logits denoting the response). Please see this for setup (Mei et al., 2023).

Now let's say that the base model attains an average pass@1 rate of $p$ on this prompt and say $n$ i.i.d. response samples drawn from the policy are used for training at one gradient step. First note that $n$ independent samples change pass@$n$ exponentially:

$$\text{pass@}n = 1 - (1-p)^n.$$

Does $n$ change the policy gradient update on the problem in one update? Averaging over $n$ samples does **not** change the expected policy gradient direction: the expected update is identical to that obtained from a single sample. What it does change is the **variance** of the gradient estimate, which decreases by a factor $n$.

Prior work (Mei et al., 2023) shows that, when using a single sample per update, tabular (stochastic) softmax policy gradient enjoys an $O(1/t)$ rate on the policy suboptimality (i.e., bound on optimal performance - attained performance) after $t$ update steps. When $n$ independent samples are used by averaging over the policy gradient update, repeating the same analysis yields

$$\mathbb{E}\Big[\text{suboptimality at step } t\Big] = O\left(\frac{A}{n \cdot t} + \frac{B}{t}\right),$$

where $B \ll A$ is a constant that does not depend on the variance of the policy gradient estimate. The constant $A$ in $\frac{A}{n \cdot t}$ depends on variance in the policy gradient estimate and corresponds to the leading term (for reasonably small $n$).

With this guarantee, the convergence rate is still linear in $t$, but the effect of stochasticity reduces drastically. For the term $\frac{A}{n \cdot t}$, $n$ and $t$ can be interchanged: one can reduce the error in this term by using a larger $n$ for a smaller $t$. The other term depends only on $t$, indicating that out of all compute allocation configurations in Section 4.1, **for instance, one should prefer the configuration that makes more sequential updates $M$ as opposed to choosing a larger $n$.** However, this is not the case in practice.

# H. A mental model of interference

A natural diagnostic is the distribution of pass@1 across prompts. Inference-time scaling laws (Schaeffer et al., 2025) relate pass@$n$ to the population pass@1 distribution, but RL training differs because the model learns from the $n$ rollouts it produces, and updates across problems introduce interference. A useful mental model is that interference is smaller when learning progress is distributed roughly uniformly across prompts. Thus, in the Fig. 25, changes in the pass@1 distribution over training can serve as a diagnostic: uniform improvement suggests controlled interference, while highly uneven improvement suggests strong interference and rich-gets-richer dynamics.

# I. More Related Works on Scaling LLM RL Axes

Prior works have explored the scaling properties of reinforcement learning (RL) for large language models (LLMs) across disparate computational axes. **On the axis of sequential scaling**, DeepSeek-R1 (Guo et al., 2025) demonstrated that RL from outcome rewards could substantially enhance reasoning capabilities. ProRL (Liu et al., 2025a) explicitly highlights the necessity of prolonged RL training. Similarly, approaches such as DAPO (Yu et al., 2025) and OpenReasonerZero (Hu et al., 2025b), though not specified, naturally extend sequential updates to until reward convergence. **On parallel rollouts per sample** (i.e., group size in GRPO), BroRL (Hu et al., 2025a) investigated the rollout width dimension, utilizing mass balance analysis to show that broadly scaling exploration guarantees correct-token mass expansion and overcomes performance plateaus observed in sequential scaling. KnapSackRL (Li et al., 2025) addressed the inefficiency of uniform sampling by formulating budget allocation as a knapsack problem, adaptively distributing resources to maximize effective gradient signals. While the impact of batch size has been extensively studied in pretraining contexts (McCandlish et al., 2018; Gray et al., 2023; Zhang et al.), to the best of our knowledge, **there is a scarcity of work systematically scaling problem batch size within the LLM RL paradigm**. Unlike previous studies that address these dimensions in isolation, our work aims to provide a unified view of compute allocation and scaling prescriptions. We note that other scaling dimensions, such as problem set size (Cheng et al., 2025), environment diversity (Zeng et al., 2025), and model parameters (Tan et al., 2025), are beyond the scope of this paper and left for future study.

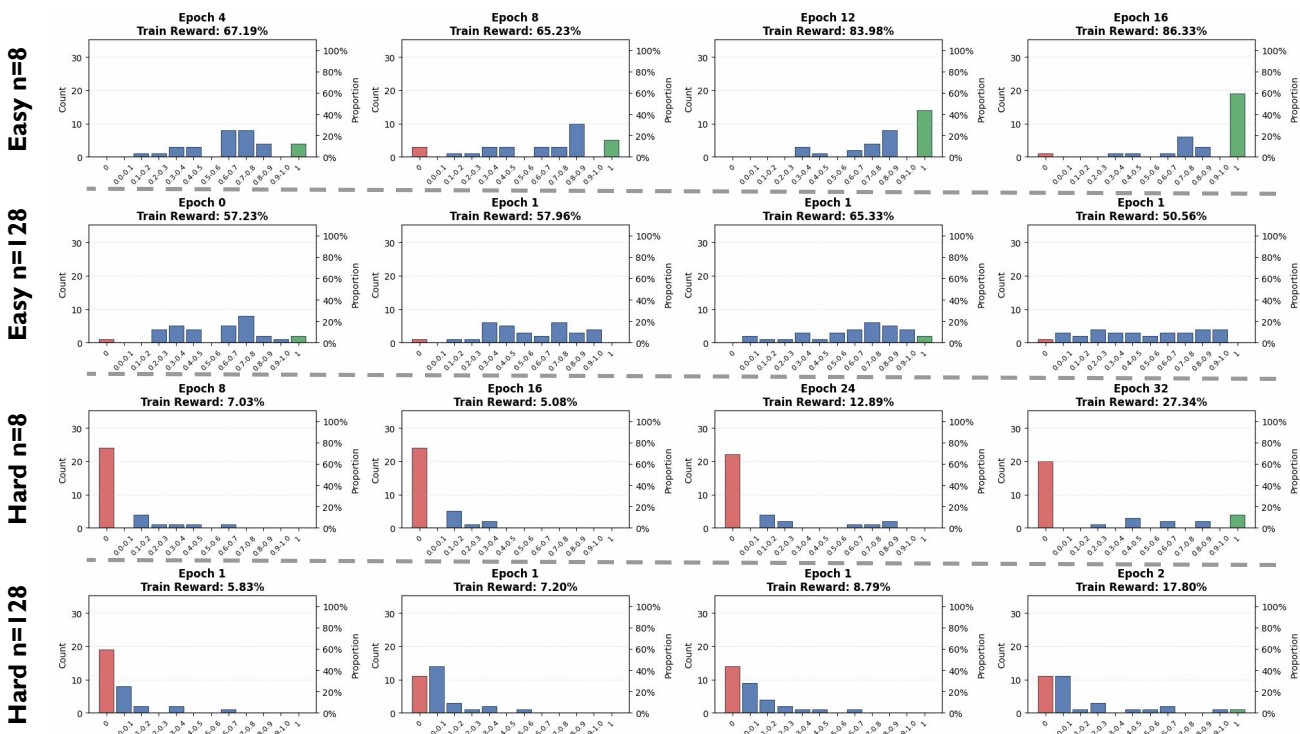

*Figure 25.* **Dynamics of pass@1 distributions (sanity-checking the interference analysis in Fig. 10).** We visualize the evolution of pass@1 histograms across training for the same four cases (Easy/Hard · $n = 8/128$) at matched compute. The temporal trajectories corroborate the main-text interpretation: on **Easy**, small $n$ progressively polarizes into a mass near 1 with a persistent non-zero fraction near 0 (optimization-induced *interference*), whereas large $n$ maintains a more dispersed, uniform distribution. On **Hard**, large $n$ increases *coverage* by steadily reducing the zero-mass, while small $n$ concentrates gains on a subset of solvable problems, yielding sharper but less comprehensive improvements.

## J. Discussion, Summary, and Future Work

A central takeaway from this work is that healthy RL recipes are inherently dependent on the prompt distribution and the behavior of RL training depends on the interaction between the base model and the prompt set, and that this dependence manifests directly in how optimal hyperparameters scale with compute. The same algorithm can exhibit qualitatively different scaling behavior on easy versus hard problem sets. On easier problems, increasing parallel rollout compute primarily improves sharpening and robustness, whereas on harder problems the dominant effect is expanded coverage. While trends in compute-optimal hyperparameters are often consistent when measured using average reward, they can diverge substantially under alternative metrics such as best@k and worst@k. This sensitivity to both data difficulty and evaluation metric highlights a key departure from supervised learning, where scaling behavior is typically more uniform once the model size is fixed. In RL, scaling laws are therefore inherently more nuanced, reflecting the coupled effects of optimization dynamics, exploration, task structure, and evaluation criteria. This study provides, perhaps the first, concrete framework for identifying and reasoning about these trends with base models and prompt sets, and empirically illustrates them across several regimes.

Our analysis also surfaces an important open challenge for future work: interference across problems. In an idealized single-problem setting, one might expect clean exponential improvements with increasing sampling compute. In practice, however, RL is performed over mixtures of problems, where progress on some tasks can interfere with learning on others. This population-level interference alters both the coefficients and effective hyperparameter values in observed scaling laws.

A promising direction is to identify sufficient statistics early in training that capture the degree of interference across problems, enabling more accurate predictions of how additional compute will translate into subsequent learning progress. We believe that tracking changes in the pass@1 distribution over the course of training provides a natural starting point for studying interference. Developing such models would be a critical step toward predictive scaling laws for RL on heterogeneous data mixtures. Mathematically, this points toward approximate closed-form rules for compute-optimal

hyperparameters that generalize across base models and prompt distributions by estimating a small number of statistics that summarize the pass@1 landscape and incorporating them into scaling-law fits. This remains an interesting direction for future work.

