# OpenReview forum: "IsoCompute Playbook: Optimally Scaling Sampling Compute for LLM RL"
_ICML.cc/2026/Conference — ICML 2026 regular_

### Official Review · Reviewer_1rGR · 2026-03-02

**Soundness:** 3
**Presentation:** 4
**Significance:** 3
**Originality:** 3
**Overall Recommendation:** 5
**Confidence:** 4

**Summary:**

This paper investigates how to optimally scale LLM reinforcement learning compute across three dimensions: parallel rollouts ($n$), problems per batch ($B_p$), and update steps ($M$). It finds that the compute-optimal number of rollouts ($n$) follows a predictable sigmoid curve as the budget increases, serving to sharpen solutions on easy problems and expand coverage on hard ones. The study provides practical rules for tuning these hyperparameters to maximize RL performance under fixed compute constraints.

**Compliance With Llm Reviewing Policy:**

Affirmed.

**Final Justification:**

The authors’ response addressed most of my concerns, so I have raised my score accordingly. I encourage the authors to incorporate the additional discussion and results into the final version of the paper. Overall, I lean toward accepting this paper because it studies an important problem and presents rigorous experimental results.

**Key Questions For Authors:**

Q1: Can the authors provide any evidence that their findings generalize to tasks(e.g. coding, general agent) outside of mathematical reasoning?

Q2: How would the optimal value of $n$ change if the "hard set" were redefined to include only problems with low but non-zero pass rates?

Q3: Is it possible or effective to dynamically scale $n$ and $B_p$(while fixing the total batch size) during the training process?

**Limitations:**

yes

**Strengths And Weaknesses:**

### Strengths:

S1: This paper investigates the optimal allocation of rollout sampling compute across parallel rollouts ($n$), batch size per prompt ($B_p$), and update iterations ($M$). This is a practical problem for improving the efficiency of RL for LLMs.

S2: The presentation of the paper is clear and easy to follow. The authors provide a detailed analysis of their empirical results and justify their experimental settings well. The conclusions are well-organized.

S3: This paper offers practical guidance for adjusting rollout-related hyperparameters in RL post-training.


### Weaknesses:


W1: The experiments are limited to mathematical reasoning tasks using the Guru-Math dataset. It is not yet clear if these scaling laws apply to other areas like coding or general agent tasks. (The weakness here is understandable, as performing experiments on a single task at this scale is already very expensive.)

W2: The finding that harder problems require a smaller $n$ is counter-intuitive, and the explanation provided in the paper is not fully persuasive. This result might be related to how the "hard set" is defined and its actual pass rate distribution. It would be useful to know if a larger n is beneficial for problems that have low but non-zero pass rates, such as those between 1/128 and 1/32.

There is a possibility that the "hard set" contains problems that are effectively unsolvable due to errors in the ground truth or incomplete task descriptions. It would be more robust if the authors could use a more powerful model with large pass@K to verify that these "hard" tasks are actually solvable.

---

> ### Author Rebuttal · Authors · 2026-03-31
>
> Thank you for the thoughtful review and for the positive assessment of the paper’s clarity, practical value, and technical quality. To address your main concerns, we added broader validation beyond math reasoning, a stratified analysis on a hard-but-solvable subset, and a preliminary test of dynamic $n$/$B_p$ scheduling. We believe these additions strengthen the paper and directly address the questions in your review.
>
> ## Generalization Beyond Math (W1 & Q1)
>
> We completely agree that demonstrating generalization beyond mathematical reasoning is crucial for validating our scaling laws, and we appreciate your understanding regarding the extreme computational cost of these experiments.
>
> During the rebuttal phase, we added experiments on larger models (32B, 70B), new tasks(code, logic, long-horizon SWE-agents), and other RL algorithms (PPO, CISPO). We compiled the new quantitative results in our response to **Reviewer 8L9w** at the top of this thread, since this question is shared across reviews. Please kindly refer to the results there. The results confirm the same qualitative phenomenon across model sizes, domains, and RL algorithms. We will update the manuscript to include these new findings.
>
> ## Solvability and the "Hard Set" Definition (W2 & Q2)
>
> Thanks for your suggestions on the hard problems. We believe unsolvable hard problems do not affect the trend. In **GRPO**, a truly unsolved problem with all-zero rewards across the $n$ rollouts contributes essentially zero group-normalized advantage, so such examples behave more like **dead compute** than like a source of misleading gradient signal.
>
> **Regarding why hard problems admit a lower saturation n value than easy problems**, our existing analysis (Section 4.1, point 4) suggests that on hard problems, larger $n$ helps **different objectives** differently. As shown in Figure 7, larger $n$ improves **best@4** more strongly, while **worst@4** saturates at much smaller $n$. We observe the same pattern on the **Extremely Hard** subset in Figure 19: larger $n$ improves best@4 at higher compute, whereas worst@4 is maximized by a moderate $n$. This suggests that larger $n$ improves **coverage / pass@k-like** behavior more than **robustness**, which helps explain why the compute-optimal $n$ on the hard set remains smaller than on the easy set.
>
> In the rebuttal phase, to test whether the original hard-set trend is caused by effectively unsolved examples, we ran a stratified analysis on a **strictly solvable hard subset** with base pass rates between $1/128$ and $1/16$ on Guru-Math with Qwen2.5-7B-Instruct. The lower bound of $1/128$ ensures that these problems are not effectively unsolved for the base model.
>
> | Pass-rate subset [1/128, 1/16] | n=8 | n=64 | n=128 | n=512 |
> |---|---:|---:|---:|---:|
> | Best performance | 16.0 | 16.7 | 17.5 | 17.5 |
> | Compute | 0.9M | 0.7M | 1M | 6M |
>
> As shown above, on this verifiable "hard-but-solvable" subset, larger $n$ values (128 or 512) still outperform smaller values like 8 or 64. The saturation point is $n=128$, the same as for the hard problems in our main study, indicating that unsolvable questions do not affect this point.
>
> ## Dynamic scaling of $n$ and $B_p$ (Q3)
>
> We believe dynamic scaling of $n$ and $B_p$ is a promising direction. While the current paper focuses on fixed allocations to derive clean scaling laws, adaptive schedules may be more effective in practice.
>
> To explore this, during the rebuttal period we are running new experiments with fixed global batch sizes of 4k and 8k and a dynamic schedule over $n$ and $B_p$. Specifically, we start with $B_p=256$, halve $B_p$ after each epoch, and increase $n$ correspondingly so that the total batch size remains constant. This schedule tests the hypothesis that broader problem coverage is more beneficial early in training, whereas allocating more samples per problem becomes more beneficial later.
>
> In our preliminary experiments, this dynamic schedule underperforms strong fixed baselines such as $(B_p,n)=(32,256)$ and $(32,512)$:
>
> | Setting | Value |
> |---------|-------|
> | Dynamic 8k | 66.3 |
> | Dynamic 4k | 64.3 |
> | Fixed 32-512 | 70.6 |
> | Fixed 32-256 | 69.8 |
>
> These results do not support the claim that this particular dynamic schedule outperforms the best fixed allocations. At the same time, they do not rule out the broader potential of adaptive allocation, since we evaluated only a single simple schedule during the rebuttal period.
>
> Our current conclusion is limited: adaptive allocation may still be promising, but likely requires more careful design than this first simple schedule. In that sense, the fixed-allocation scaling laws studied in this paper remain a useful and competitive baseline for compute allocation under a fixed budget.
>
> Thank you again for the constructive feedback. We hope these additional experiments and clarifications help address your concerns.

---

> > ### Author Rebuttal · Reviewer_1rGR · 2026-04-02
> >
> > I appreciate the authors' responses. Most of my concerns have been addressed, and I remain in favor of accepting the paper.

---

> > > ### Author Response · Authors · 2026-04-03
> > >
> > > Thank you again for your thoughtful engagement and for taking the time to carefully consider our rebuttal. We are very glad to hear that your concerns have been adequately addressed. We sincerely appreciate your feedback, which we believe has helped improve the paper. If you feel the rebuttal and planned revisions have resolved the key issues you raised, we would be very grateful if you could consider raising your score.

---

### Official Review · Reviewer_ZBjA · 2026-03-11

**Soundness:** 3
**Presentation:** 3
**Significance:** 2
**Originality:** 2
**Overall Recommendation:** 4
**Confidence:** 3

**Summary:**

This paper targets discovering scaling laws for post-finetuning of LLM with GRPO on math tasks. They consider 3 different compute dimensions: $B_p$ the number of unique prompts per batch, $n$ number of rollouts for the same prompt in a batch and $M$ and number of gradient update steps. For the fixed compute budget $C = n B_p M$, they explore several different scenarios on how does the model performs: $n$ vs $M$ to measure the importance of data collection vs update steps; $n$ vs $B_p$ to understand whether more diverse prompts or more rollouts per prompt is more important in one update; finally how to jointly choose $M$, $n$, $B_p$. Their results emphasise the importance of scaling $n$ and its eventual saturation following the sigmoidal law.

**Compliance With Llm Reviewing Policy:**

Affirmed.

**Final Justification:**

I have increased the soundness and presentation score, which results in weak accept.
Overall, the authors have addressed almost all my concerns and for remaining concerns, they promised to update the final version to address them.

**Key Questions For Authors:**

- L360: "pick the smallest stable $B_p$" -> what does it mean, stable $B_p$? Can you provide more suggestions on how to choose $B_p$
- Why haven’t you tested $B_p$ vs $M$?
- Is your research only valid for finetuning with GRPO?
- Have you tested on slightly "non-healthy RL" scenarios?
- Why different levels of frontiers significantly overlap in terms of compute budget in Figures 6 and 8?

**Limitations:**

yes

**Strengths And Weaknesses:**

## Strengths

- The authors bring new insights that scaling $n$ is more beneficial over scaling $B_p$ for both easy and hard tasks
- Discovering the phenomenon of saturation of $n$
- They provide an empirical recipe for compute budget partioning.
- Explore separately the question of scaling for "easy" and "hard" problems

## Weaknesses

 - For a fully empirical and experimental paper, I find that the authors didn’t test enough their hypotheses on different base models (they have only considered 2 different base models). While redoing such large scale experiments for all base models is prohibitively expensive, I find it still important to revalidate the hypothesis of $n^*(C)$ being monotonically increasing and upper bounded on very diverse set of models of different sizes and capabilities as **it is positioned as one of the main discovery of the paper**. This could be done by considering a smaller range $n$ for 2-3 fixed values of $B_p$. The same could be done for different math datasets, like AIME 2024 - 2025.
- I am not fully convinced by the saturation phenomenon: can it be that the increase of $n^*(C)$ just slows down but can be still unlimited? Intuitively, bigger $n$ should not cause any damage and, on the opposite, should increase the statistical reliability and the total batch size per update. Is it possible, that we just need to consider much larger $C$ to see an increasing effect again. Or the learning rate is not sufficiently adapted for larger batch sizes? Even from the plots, e.g. Figure 6 and 8, it is not obvious that the increase is capped, at least on easy tasks.
- Fitted laws for $n^*(C)$ vary quite a lot on all the parameters from one plot to another; in general no statistical errors are provided to understand how well the sigmoidal law fits the frontiers. Do you fit all the frontier points we see on the Figure 6? If yes, then I would say it is better to fit the curve only to the leftmost point in each level, the remaining points are there because intermediate values of $n$ are not available.
- While the authors mention that "However, on the Hard set or skewed problem distribution, we speculate they may require a higher minimum $B_p$ for effective training." This thought is not developed, in general, while $B_p$ seems to be less important than $n$, its role might be slightly undermined by the authors, especially for the hard tasks as Figure 9 suggests.
- In general, this work seems to be too limited to one RL scenario, where one uses GRPO to fine-tune LLM on math tasks. It doesn’t cover other RL techniques or reasoning tasks where instead of a single output per prompt, you generate a chain-of-thoughts.

---

> ### Author Rebuttal · Authors · 2026-03-31
>
> We thank Reviewer ZBjA for their thoughtful feedback and for recognizing the value of our empirical recipe and the novelty of our findings regarding $n$-scaling. Below we address the main concerns and summarize additional experiments.
>
> ## Generalizability across Models, Tasks, Algorithms, and CoT (Weakness 1, 5 & Q3)
>
> Thank you for your interest in generalization of our findings. We would first like to clarify that our main finding is well supported by not two, but **three** models in the paper: Qwen2.5-7B-Instruct, Qwen3-4B-Instruct, and Llama-3.1-8B-Instruct in Figure 5 and Figure 11.
>
> New experiments during rebuttal on larger models (32B, 70B), new tasks (code, logic, SWE-agents), and other RL algorithms (PPO, CISPO) confirm the same trend, summarized in our response to **Reviewer 8L9w**. Note it is hard to find the exact saturation point with sparsely sampled values of $n$ within this rebuttal period.
>
> Chain-of-Thought (CoT) and AIME benchmark. We clarify that **all** experiments use long CoT (8k max response tokens). AIME2024 OOD results in Appendix C confirm larger $n$ outperforms smaller $n$.
>
> ## The Saturation Phenomenon and Learning Rate (Weakness 2)
>
> Thanks for the question. Figure 6 and 8 already show the maximum $n$ we tested (the horizontal dashed lines): on the easy set, we ran up to $n=2^{11}$ and $n=2^9$ for hard set, with saturation observed at $n=2^9$ and $n=2^7$. We believe this provides clear evidence that the saturation phenomenon is genuine.
>
> Regarding why this saturation happens, we agree this saturation may seem counterintuitive at first, since a larger $n$ can improve the gradient quality for a given problem. In fact, we see larger $n$ results in a higher reward at large compute budgets on **the train set** (Figure 11, left; Section 5.2 “train-test gap”). However, the scaling laws are defined by validation reward, which may diverge from the training reward with larger $n$.
>
> Regarding the learning rate (LR), as noted in Section 2, we ablated sqrt-root LR scaling with batch size, which we found effective. Besides, the sqrt-root LR scaling was commonly used when training with large batch size ([Hoffer etc.](https://arxiv.org/pdf/1705.08741)). We believe our LR choice should be sound.
>
> ## Curve Fitting and Frontiers (Weakness 3 & Q5)
>
> Regarding the curve fitting, the sigmoid curves are fitted using all the points. As detailed in Section 4 (“Data Analysis Workflow”) and Figure 14, we define the frontier as the “record-breaking points”.
>
> Regarding the overlapped frontier $n$ on Figure 6 and 8, it is a consequence of RL variance: a larger $n$ may set a new record first, while a smaller $n$ can occasionally surpass it later. This reflects local stochasticity, while we show the overall frontier still shifts toward larger $n$ as compute increases.
>
> To show how well the sigmoidal fits the frontier sets, we evaluated fit quality on the **same fit sets in the main-paper figures**. Sigmoid achieves the lowest error compared to others (e.g. linear(power), saturating exponential fit(sat_exp), and log).
>
> | Model | SSE (Fig.6 / Fig.8) | RMSE (Fig.6 / Fig.8) | MAE (Fig.6 / Fig.8) |
> |---|---|---|---|
> | sigmoid | 8.62 / 21.62 | 0.75 / 0.52 | 0.45 / 0.33 |
> | sat_exp | 11.99 / 23.41 | 0.89 / 0.55 | 0.71 / 0.38 |
> | linear(power) | 12.51 / — | 0.91 / — | 0.74 / — |
> | log_M | — / 34.91 | — / 0.67 | — / 0.54 |
>
> Regarding the leftmost-only suggestion, we also tested it. The best fits become linear, but leftmost-only discards plateau segments where the frontier-optimal $n$ remains unchanged over a wide compute range. These non-leftmost record-breaking points are exactly the evidence for flattening, so we maintain the full set of record-breaking points.
>
> ## The Role of $B_p$ and Trade-offs with $M$ (Weakness 4, Q1, Q2)
>
> Regarding the $B_p$ effects on hard problems, we observed $B_p$ has a larger effect on hard problems than easy ones (Figure 9, line 315). We do not quantify this mechanism here and leave it for future work.
>
> By "smallest stable" $B_p$ (Q1), we mean the minimum $B_p$ that yields consistent reward gains without catastrophic variance. In our sweeps, this was $B_p=32$. In practice, one can decrease $B_p$ until training destabilizes.
>
> Regarding $B_p$ vs $M$ (Q2): Figure 9 isolates the effect of $B_p$ by fixing $n$. We didn’t explicitly plot $M$ because we did not observe a clear relation of optimal $M$ on $B_p$. We also discussed the joint optimization of $(B_p, n, M)$ in Section 4.3, where the $B_p$ vs $M$ view can be extracted from the frontier.
>
> ## Non-Healthy RL Scenarios (Q4)
>
> We believe it is not fruitful to identify scaling laws where training is unstable, as no predictable allocation trends can be made when learning is itself unstable. Even in pretraining, scaling laws can’t be derived from (and existing scaling laws do not apply to) settings where the learning rate is too high or gradients blow up. Therefore, we omit non-healthy RL scenarios.
>
> Thank you again for the constructive feedback.

---

> > ### Author Rebuttal · Reviewer_ZBjA · 2026-04-04
> >
> > Thank you for thoughtful replies. Overall, your answers have well addressed most of my concerns. About saturation phenomenon, your explanation in the rebuttal made it more clear, I believe section 5.2 is very important for understanding this phenomenon and I suggest emphasizing more the effect of fixed $D$ on the saturation of $n$
> >
> > Some remaining concerns:
> > -In Figure 11, overfitting for larger $n$ is not confirmed on Qwen3-4B-Instruct (hard problem). Does it mean that this case may require larger $n$ to saturate?
> > - After re-reading section 5.2, I wonder if prompt size $D$ should not enter as one of the component of the computational budget $C$. Indeed, the larger prompt implies more expensive inference of the models. Moreover, Figure 12 suggests a non-trivial relationship between $n^*$ and $D$, which is worth studying.
> > -  I would like to comment further on the statement "Regarding the leftmost-only suggestion, we also tested it. The best fits become linear, but leftmost-only discards plateau segments where the frontier-optimal $n$ remains unchanged over a wide compute range. These non-leftmost record-breaking points are exactly the evidence for flattening, so we maintain the full set of record-breaking points.". The reason why you need to consider "non-leftmost" points is because you don't have a large enough set of possible $n$.  If you consider a larger set of possible $n$, then we can expect the smoother transition in $n^* (C)$ , which would give us a true scaling law. Computing a scaling law on non left-most points, the way you do, may distort the true $n^* (C)$. How do you make sure that this distortion is limited?

---

> > > ### Author Response · Authors · 2026-04-05
> > >
> > > Thank you for your follow-up questions and we are glad that our initial response addressed most of your concerns. We address your remaining points below:
> > >
> > > **Regarding the saturation on Qwen3-4B-Instruct (Hard Problems)**, you’re right our evaluations up to $n=64$ indeed show monotonic performance gains. The actual saturation point is likely to be higher under this setup. We will experiment with larger n values ($n\geq128$) for the final version.
> > >
> > > **Regarding prompt set size $D$**, we believe there might be a misunderstanding here: by $D$ we do not refer to the length of the prompt, as in the number of tokens in the prompt. Rather we use $D$ to refer to the **number of prompts (problems)** in the training dataset.  Under our scaling study setup, the size of the prompt in terms of number of tokens does not have an effect on $C$ as $C \propto B_p * n * M$ and we used a fixed D (6000 easy problems, 5000 hard problems) in the main scaling law study  (Section 3 and 4).
> > >
> > > In our rebuttal response, we indicated that size of the prompt set $D$ affects the saturation value of $n^{\*}$ because training on a smaller dataset (fewer prompts, smaller $D$) will result in overfitting such that gains on training performance may not translate to test set anymore. Therefore, a smaller $D$ will result in a smaller value of the saturation value of $n^{\*}$ (line 433). This relationship is also precisely shown in Figure 12, where we find that the saturation value of $n$ as we scale compute is smaller when $D = 500$ compared to when $D = 6000$.
> > >
> > > **Regarding leftmost-only suggestion**, we agree with your intuition: if we had a continuous and infinitely dense set of sampled $n$ values, the transition in $n^*$ would be perfectly smooth, and fitting only the leftmost points would indeed yield the true, undistorted scaling law. However, in the practical workflow to study the scaling law,
> > > + **Denser sampling of $n$ will reduce distortion:** As we sample $n$ more densely, we expect the curves from leftmost fitting and all-points fitting to become closer. The all-points fitting data has a staircase shape, but the width of each step should shrink as we sample $n$ values more densely over the range we study. So the amount of deviation between scaling curves from leftmost fitting and all-points fitting should be limited and should decrease as we sample more $n$ values.
> > > + **Maximum value of $n$ is not the largest value tested because of the train-test gap:** We often observe that increasing $n$ eventually hurts validation performance when the number of training prompts is fixed. As a result, the leftmost point need not correspond to the largest tested value of $n$, since larger $n$ can suffer from a train-test gap. If the dataset size were also allowed to grow, this saturation would likely disappear; but under a fixed dataset size, it persists. This further supports our conclusion that saturation appears regardless of whether we use leftmost fitting or all-points fitting.
> > > + **Practical limitations:** Due to compute limitations, we can't scale $n$ infinitely in our experiments, like scaling $n$ values to be both larger and denser. In addition, practitioners are unable to run larger $n$ value beyond a maximum due to hardware limitations. Therefore a law based on a maximum value of $n$ practically is sensible.
> > >
> > > That said, we agree that we should do a dense study within the range we can -- and we will do it for the final, but we believe that the key takeaways of the analysis would not change for a practitioner and per our argument above, a larger number of values of $n$ will only result in similar scaling curves with both fitting strategies..
> > >
> > > Please let us know if this response addresses your concern and we are happy to discuss further. We would be deeply grateful if you might consider raising your score if your primary concerns have been addressed.
> > >
> > > ---
> > > **Update**: As today is the last day of the author-reviewer discussion period, we would greatly appreciate it if you could let us know whether our latest response resolves your remaining questions. If so, we kindly ask you to consider updating your score to reflect the progress made during this discussion. We are also happy to address any further concerns you may have.
> > > Thank you for your time and effort in reviewing our work.

---

### Official Review · Reviewer_8L9w · 2026-03-13

**Soundness:** 3
**Presentation:** 3
**Significance:** 3
**Originality:** 2
**Overall Recommendation:** 4
**Confidence:** 4

**Summary:**

This paper investigates the optimal allocation of sampling computation for on-policy reinforcement learning methods in large language models. It emphasizes the concept of scaling as a compute-constrained optimization problem, considering three key resources: the number of parallel rollouts per problem, the number of problems per batch, and the number of update steps. This work contributes to a deeper understanding of scaling laws in LLM reinforcement learning.

**Compliance With Llm Reviewing Policy:**

Affirmed.

**Final Justification:**

The authors have responded very thoroughly to my main concerns, especially by providing ample code and experiment logs, which gives me further reasons to support them. Therefore, I have decided to raise my scores for Soundness (2 -> 3) and Overall Recommendation (3 -> 4).

**Key Questions For Authors:**

1. See Weakness
2. Could you provide further theoretical analysis based on the experimental conclusions?

**Limitations:**

1. The author should provide further clarification regarding reproducibility.

**Strengths And Weaknesses:**

Strengths：

1. This paper provides a relatively comprehensive summary of the research on scaling laws in LLM reinforcement learning, offering a fairly complete perspective.
2. The author conducts a large number of experiments, and the experiments generally align with the author's conclusions.
3. The experimental results are well visualized, and the images are exquisite.

Weaknesses：

1. The primary contribution of the paper is its experimental work. Unfortunately, the authors do not provide the relevant experimental code for review. If the authors could supply the code necessary to reproduce the experimental results, it would enhance the reliability of the paper's conclusions.
2. While empirical findings are presented, translating these results to larger, even more complex models or differing computational resources may pose challenges. I believe it is necessary to verify whether similar conclusions hold true on larger-scale models. Otherwise, drawing inferences about its applicability to larger models based solely on the current experimental results would be rather premature.

---

> ### Author Rebuttal · Authors · 2026-03-31
>
> Thanks a lot for the review! We are glad that you feel we did a lot of experiments and the paper is complete. To address your concerns regarding experimental work, we are releasing the code in an anonymous repository along with the wandb logs (identity information redacted) trained with 120K GPU hours. This should help the community reproduce results and refer to existing runs for deriving any more conclusions if needed. The code and experiment logs are here https://anonymous.4open.science/r/isoCompute-F823. Please let us know if there are any remaining concerns.
>
> **Regarding the generalization of our findings**, the original submission already shows that our main finding that optimal n scales with compute transfers across **Qwen2.5-7B-Instruct, Qwen3-4B-Instruct, and Llama-3.1-8B-Instruct**.
>
> In the rebuttal period, we added a series of new experiments on generalization: **larger models (32B, 70B)**, **new domains (algorithmic coding, logic puzzle, long-horizon SWE agent)**, and **additional RL algorithms ([PPO](https://arxiv.org/abs/1707.06347), [CISPO](https://arxiv.org/abs/2506.13585))**.
>
> For **Code / Logic**, we use the domains from [Guru](https://arxiv.org/abs/2506.14965), sampling **1K training examples** and evaluating on **4 held-out splits of 200 examples each**. For [SWE agent](https://arxiv.org/abs/2310.06770), we evaluate an **agentic software-engineering setting** on [R2E-Gym](https://arxiv.org/abs/2504.07164) using **Qwen3-32B**, with a maximum of **50 turns** per episode and 64k max length. For the largest model **K2-V2 70B**, we trained on Guru-Math data with 64k max length. Each table cell below reports the **best reward achieved**, with the **compute required to reach that point** in parentheses.
>
> | Model | Task | n=8 | n=64 | n=128 |
> |------|------|-----|------|-------|
> | Qwen2.5-7B-Instruct | Guru-Code | 68.7 (0.2M) | 69.8 (0.4M) | **73.9** (0.5M) |
> | Qwen2.5-7B-Instruct | Guru-Logic | 30.7 (770K) | **31.9** (2.7M) | 31.3 (4M) |
> | Qwen3-32B | SWE agent (max 50 turns, R2E-Gym data) | 35.9 (118K) | **43.5** (147K) | - |
> | K2-V2 70B | Guru-Math | 85.0 (389K) | **87.1** (1.3M) | - |
>
> Across these settings, we observe the same qualitative trend as in the main paper: at higher compute budgets, larger $n$ becomes preferable, with saturation at sufficiently large $n$. This suggests that the main phenomenon is **not confined to Math or to smaller model scales**, although we agree that the scope of this conclusion should be stated carefully. Note it is hard to find the exact saturation point with experiments conducted over sparsely sampled values of $n$ within this rebuttal period.
>
> Below are the experiments with PPO and CISPO algorithms. The train and validation data are Guru-Math and the model is Qwen2.5-7B-Instruct – the same configuration as the GRPO experiments in the paper.
>
> | Algorithm | Data | n=32 | n=64 | n=128 | n=256 |
> |----------|---------|------|------|-------|-------|
> | PPO | Easy | 58.8 (2.6M) | 62.3 (2.3M) | **64.8** (2.7M) | 64.3 (2.8M) |
> | PPO | Hard | 15.7 (1.1M) | 17.1 (2.2M) | **17.9** (1.8M) | 17.0 (2.1M) |
> | CISPO | Easy | 61.3 (3.1M) | - | - | **69.4** (4.6M) |
>
> These results show the same qualitative pattern beyond GRPO. We will revise the paper to make the scope of this claim explicit: the contribution is an **empirical law within the tested regimes**, not a claim of universal extrapolation to all models and settings.
>
> **Regarding further theoretical analysis**, we are happy to provide theoretical results, but could you please elaborate on what kinds of results will be useful to show theoretically? We believe we showcase a simple result extending prior work from multi-armed bandits to underscore the differences between best allocations of parallel and sequential compute in a tabular setting and an LLM setting (see Appendix G), and were wondering if there was a particular result beyond this that would be helpful.
>
> Thank you again for the constructive feedback. We believe these additions and clarifications substantially strengthen the paper and address the concerns raised in your review. If these changes resolve the main issues, we would greatly appreciate reconsideration of the score.

---

> > ### Author Rebuttal · Reviewer_8L9w · 2026-04-04
> >
> > Thanks for the authors’ detailed rebuttal. The authors have responded very thoroughly to my main concerns, especially by providing ample code and experiment logs, which gives me further reasons to support them. Therefore, I have decided to raise my scores for Soundness (2 -> 3) and Overall Recommendation (3 -> 4).
> >
> > Regarding the theoretical analysis question I raised, using multi-armed bandits for the analysis—does this assumption seem too strong? Could you please elaborate on the reasoning behind doing this?

---

> > > ### Author Response · Authors · 2026-04-05
> > >
> > > Thanks for engaging with us and for increasing your score! We are glad that your main concerns are addressed.
> > >
> > > Regarding the theoretical analysis question, we use a tabular multi-armed bandit setup in Appendix G not because we believe LLM post-training can be well modeled as a bandit problem. LLM training fundamentally involves function approximation and generalization from a pretrained model, neither of which is captured by a tabular bandit. Instead, we use the bandit analysis as a deliberately simple reference point for building intuition about why our empirical scaling results are surprising relative to the closest standard theoretical framework.
> > >
> > > Our motivation for using this framework is that a multi-armed bandit still captures important structural features of the policy: it is a softmax categorical distribution over a discrete action space, which matches the basic functional form of an autoregressive language model. It captures the nuance of training such a policy with an outcome reward signal. This makes it a useful theoretical abstraction, even though it omits the shared representation and cross-prompt generalization in LLMs. The point of the analysis is therefore not to faithfully model LLM training, but to clarify what predictions we would expect from a classical setting and then contrast those predictions with what we actually observe.
> > >
> > > In particular, as we discuss in Appendix G, the bandit analysis suggests that increasing sequential compute $M$ should be more valuable than increasing the number of parallel rollouts $n$, since in a tabular setting each arm is represented independently and there is no meaningful sharing across updates. Empirically, however, this is not what we observe. In practice with LLMs, scaling the number of parallel rollouts is more important than scaling sequential updates. We view this gap as evidence that parameter sharing in LLMs qualitatively changes the optimization picture: additional rollouts can improve learning not only for a single prompt, but across prompts to address interference (see Appendix H). Therefore, we believe this theoretical analysis is still valuable.
> > >
> > > We are happy to incorporate theoretical analysis to study the LLM setting, but we are not sure if you had a particular mental model or abstraction to get started on studying it. Therefore in our rebuttal, we kindly requested you for suggestions regarding the same.
> > >
> > > Please let us know if this response addresses your concern; if so, we would be grateful if you are willing to raise your score any further. Thank you so much!

---

### Decision · Program_Chairs · 2026-04-30

**Decision:**

Accept (regular)

**Comment:**

This paper provides highly practical rules for how to best use computing power when training LLMs with RL. The authors show that generating more practice responses per problem improves learning as overall budget grows, but this benefit eventually hits a ceiling to prevent the model from simply memorizing the training data. All reviewers recommend acceptance because the authors provided a thorough rebuttal with new experiments on larger models, proving their guidelines are reliable and valuable for the AI community.